# Exploring 3D Dataset Pruning

Xiaohan Zhao [1]   Xinyi Shang [1 2]   Jiacheng Liu [1]   Zhiqiang Shen [1]

## Abstract

Dataset pruning has been widely studied for 2D images to remove redundancy and accelerate training, while particular pruning methods for 3D data remain largely unexplored. In this work, we study dataset pruning for 3D data, where its observed common long-tail class distribution nature make optimization under conventional evaluation metrics *Overall Accuracy (OA)* and *Mean Accuracy (mAcc)* inherently conflicting, and further make pruning particularly challenging. To address this, we formulate pruning as approximating the full-data expected risk with a weighted subset, which reveals two key errors: *coverage error* from insufficient representativeness and *prior-mismatch bias* from inconsistency between subset-induced class weights and target metrics. We propose *representation-aware subset selection* with per-class retention quotas for long-tail coverage, and *prior-invariant teacher supervision* using calibrated soft labels and embedding-geometry distillation. The retention quota also serves as a switch to control the *OA-mAcc* trade-off. Extensive experiments on 3D datasets show that our method can improve both metrics across multiple settings while adapting to different downstream preferences. Code is available at https://github.com/XiaohanZhao123/3D-Dataset-Pruning.

## 1. Introduction

Dataset pruning ([Paul et al., 2021a](#); [Sener & Savarese, 2017a](#)), also termed coreset selection, is a mature strategy for reducing redundancy in large-scale training sets. In 2D image classification, a broad family of methods has

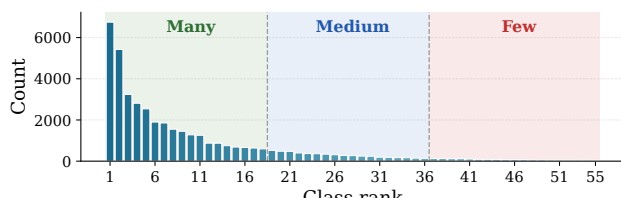

*Figure 1.* Grouped class distribution of ShapeNet55.

*Table 1.* Statistics of 3D imbalance on *both* train and test set.

| Dataset | Train Set | | | | Test Set | | | |
|---|---|---|---|---|---|---|---|---|
| | Total | Max | Min | Ratio | Total | Max | Min | Ratio |
| ScanObjectNN | 11,416 | 1,585 | 298 | 5.3× | 2,882 | 390 | 83 | 4.7× |
| ModelNet40 | 9,840 | 889 | 64 | 13.9× | 2,468 | 100 | 20 | 5.0× |
| ShapeNet55 | 41,952 | 6,747 | 44 | 153.3× | 10,518 | 1,687 | 12 | 140.6× |

been validated, including geometry-based clustering ([Sinha et al., 2019](#)), gradient matching ([Mirzasoleiman et al., 2020](#); [Killamsetty et al., 2021](#)), and error-based importance metrics ([Paul et al., 2021a](#); [Toneva et al., 2018](#)). These approaches accelerate iteration and reduce training cost, and have been adopted in settings such as hyper-parameter tuning ([Yang et al., 2022](#)) and continual learning ([Hao et al., 2023](#)). More recently, pruning has been combined with knowledge distillation ([Chen et al., 2025](#); [Ben-Baruch et al., 2024](#)), where a teacher trained on the full dataset guides both subset selection and post-pruning training.

Despite this progress, dataset pruning remains under-explored for 3D modalities, where data are expensive to curate and class frequencies are rarely controlled. 3D datasets are typically built via manual modeling (e.g., CAD repositories) or real-world scanning, and therefore reflect natural object frequencies or downstream-driven modeling preferences. As a result, long-tailed imbalance often appears in *both* training and test splits (Fig. 1, Tab. 1).

This 3D setting brings a practical dilemma between two widely used metrics. Mean class accuracy (mAcc) measures balanced generalization across classes, while overall accuracy (OA) reflects practical utility under the naturally imbalanced evaluation prior. Importantly, in 3D classification benchmarks OA should not be dismissed as a biased metric: when the test distribution is long-tailed, OA is the empirical estimate of expected accuracy for real queries. Therefore, pruning in 3D should respect both, rather than treating OA as merely an inferior alternative to mAcc.

We argue that this dilemma should not be addressed by a

[1]VILA Lab, Department of Machine Learning, MBZUAI [2]Department of Statistical Science, University College London. Correspondence to: Zhiqiang Shen <zhiqiang.shen@mbzuai.ac.ae>.

*Proceedings of the $43^{rd}$ International Conference on Machine Learning*, Seoul, South Korea. PMLR 306, 2026. Copyright 2026 by the author(s).

premature trade-off. Instead, the key is to first establish principles that remain *robust and beneficial* under diverse target priors, and then support preference steering on top of such a shared foundation. To make this concrete, we cast pruning as a quadrature approximation of population risk and decompose the learning error into *representation error* (fidelity to the underlying manifold) and *prior-mismatch bias* (distribution shift induced by the subset). This view validates that no single subset fits all target priors, yet it also provides direct guidance: it suggests which components can be strengthened in a prior-robust manner, before adapting to specific evaluation preferences.

Guided by this theoretical decomposition, our framework targets both components. To reduce *prior-mismatch bias*, we decouple a prior-robust structural likelihood from a prior-dependent offset. We realize this via KD, which provides instance-wise soft targets and makes learning less sensitive to prior rebalancing. We further refine this structural signal with teacher calibration and geometry- and relation-preserving distillation. To reduce *representation error*, we audit selection signals under imbalanced 3D data and find that classifier-derived scalar scores strongly correlate with class size, whereas embedding geometry is more stable (with stronger inductive bias). We therefore select for coverage in the embedding space. Our analysis also motivates a small per-class safety floor to secure the shared high-return regime across priors. Finally, since no single subset is optimal for all priors, we introduce a simple steering wrapper that interpolates between stratified seeding and global selection, enabling preference steering between mAcc and OA. We call the resulting framework `3D-Pruner`.

Our contributions are summarized as follows:
**(i)** We highlight a 3D pruning dilemma under long-tailed train/test splits, where OA and mAcc correspond to distinct, practically meaningful evaluation priors.
**(ii)** We cast pruning as quadrature over population risk and decompose the error into representation error and prior-mismatch bias, yielding prior-robust principles shared across target priors.
**(iii)** We propose `3D-Pruner` based on these principles, which elevates the performance floor across priors while enabling flexible preference adjustment between mAcc and OA via a simple steering wrapper. To our best knowledge, this is the *first* principled exploration of 3D dataset pruning.

## 2. Background

**3D Datasets.** 3D recognition benchmarks are essential for geometric learning, yet such data are costly to acquire and curate. Widely used datasets derive from two pipelines: curated CAD repositories (e.g., ModelNet40, ShapeNet (Wu et al., 2015; Chang et al., 2015; Yi et al., 2016)) and real-world scanning with depth sensors (e.g., ScanObjectNN (Uy

et al., 2019)). Both share a common property: class frequencies are shaped by object availability and annotation cost rather than deliberate control, leading to long-tail imbalance in both training and evaluation splits (Fig. 1, Tab. 1).

This setting motivates reporting two complementary metrics driven by different *evaluation priors*. Let $C$ be the number of classes, $N$ the total test examples, and $n_y$ the test examples in class $y$. Define per-class accuracy $\mathrm{Acc}_y := \frac{1}{n_y} \sum_{i:y_i=y} \mathbf{1}\{\hat{y}_i = y_i\}$. Then:
***Overall accuracy (OA)*** with empirical prior $\pi_y^{\mathrm{emp}} = n_y/N$:

$$\mathrm{OA} = \sum_y \pi_y^{\mathrm{emp}} \, \mathrm{Acc}_y.$$

OA reflects *frequency as utility*: users predominantly query common objects (e.g., doors, tables), so high OA indicates reliability for daily use with presumably higher tolerance of rare cases like a strange ancient vase, where the system's value is determined by the cumulative success rate of user interactions.
***Mean class accuracy (mAcc)*** with uniform prior $\pi_y^{\mathrm{uni}} = \frac{1}{C}$:

$$\mathrm{mAcc} = \sum_y \pi_y^{\mathrm{uni}} \, \mathrm{Acc}_y.$$

mAcc reflects *capability versatility*: every category is equally important regardless of prevalence. High mAcc certifies discriminative features for all defined concepts, not merely exploiting frequency priors. Importantly, OA here should not be conflated with dense prediction settings where a dominant "background" label inflates overall scores. In 3D classification benchmarks, head classes correspond to real object categories that appear more often in scans or are more frequently modeled to meet market preference, yet still exhibit substantial intra-class variation. Thus, 3D models routinely report both metrics as equally legitimate but differently biased targets (Qi et al., 2017; Qian et al., 2022; Deng et al., 2023). This duality is central for pruning, since subset selection can implicitly change the effective class prior and shift the balance between these two objectives.

**Data Pruning.** Dataset pruning, also termed as coreset selection, aims to identify a compact subset of training examples that preserves most of the learning signal of the full dataset, thereby reducing training cost. Existing methods span coverage and diversity-based selection, such as herding, k-center style objectives, and submodular subset selection (Welling, 2009; Sener & Savarese, 2017b; Wei et al., 2015), as well as gradient-based matching that approximates the optimization trajectory induced by the full data (Mirzasoleiman et al., 2020; Killamsetty et al., 2021). Another line of work ranks examples using training dynamics or error statistics and prunes those deemed redundant or uninformative (Toneva et al., 2018; Paul et al., 2021b). Recently, teacher signals from knowledge distillation (Hinton

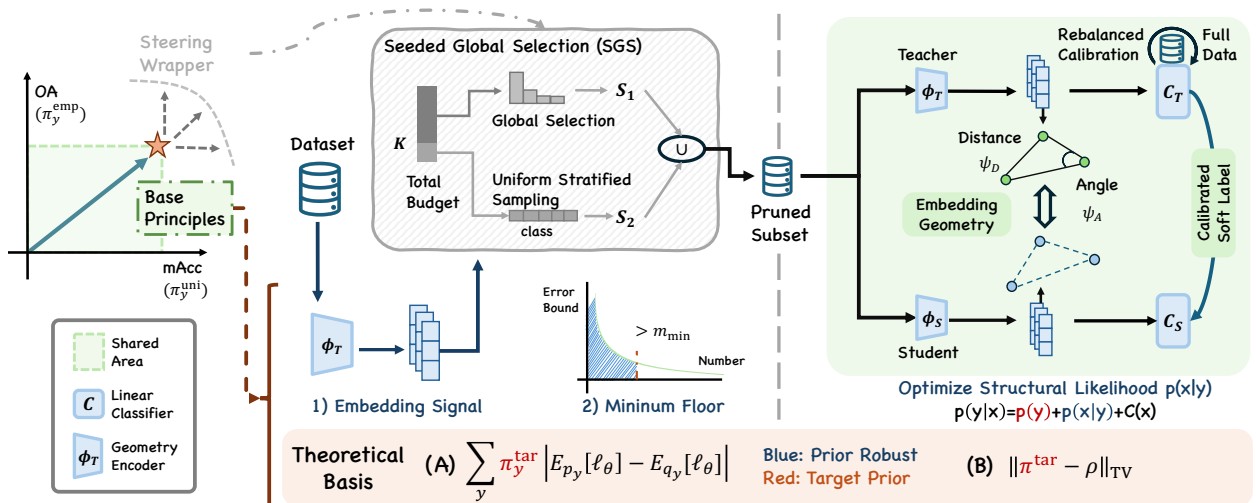

*Figure 2.* Illustration of our `3D-Pruner` framework, which comprises: (1) base principles (utilizing embedding signals, minimum floor selection, and optimizing structural likelihood in post-pruning training) that remain robust and beneficial across different priors, derived from theoretical analysis of the shared region, and (2) a steering wrapper that balances between the two priors.

et al., 2015), especially soft labels, have also been incorporated into data pruning and subset training, improving the robustness and performance of pruned learning (Hinton et al., 2015; Ben-Baruch et al., 2024; Chen et al., 2025). While prior studies are largely centered on 2D benchmarks, pruning in 3D often faces class imbalance, where the effect of pruning depends on the evaluation prior. Several recent methods address imbalance-aware pruning by enforcing class-wise stratified sampling with class-quota estimated by class-wise difficulty score (through a hold-out set performance or class-wise aggregation on scalar importance values like EL2N), but typically optimizing a fixed target criterion such as mean per-class accuracy or worst-class accuracy (Vysogorets et al., 2024; Zhang et al., 2025; Tsai et al., 2025). In contrast, we explicitly study pruning under two priors, the natural distribution and the mean-class distribution, and explicitly characterize their confidant and overlap in the 3D setting.

## 3. Theoretical Analysis

To characterize the conflict between OA and mAcc, we formalize dataset pruning as a quadrature approximation of the population risk. This formulation enables an explicit decomposition of learning error into representation-driven and prior-driven components, thereby exposing the mathematical origin of this conflict.

**Pruning as quadrature approximation.** Let $p$ denote the target distribution over examples $x$ (for brevity, $x$ may include an input–label pair), and define the population risk $\mathcal{L}(\theta) = \mathbb{E}_{x \sim p}[\ell_\theta(x)]$. Given a subset $S \subset \{1, \dots, n\}$ of size $m$ and weights $w \in \Delta_m := \{w_i \geq 0, \sum_{i \in S} w_i = 1\}$, define the discrete measure $q_{S,w} = \sum_{i \in S} w_i \delta_{x_i}$ with $\delta$

denoting the Dirac function. Therefore, we can think of pruning as using a quadrature rule to approximate such integration, under the following form:

$$\hat{\mathcal{L}}_{S,w}(\theta) = \mathbb{E}_{x \sim q_{S,w}}[\ell_\theta(x)] = \sum_{i \in S} w_i \ell_\theta(x_i). \quad (1)$$

The following lemma links the quality of this approximation directly to the generalization gap of the trained model.

**Lemma 3.1** (Generalization gap via discrepancy)**.** *Let* $\mathcal{G} = \{\ell_\theta(\cdot) : \theta \in \Theta\}$ *and define the discrepancy* $D_\mathcal{G}(p, q) = \sup_{g \in \mathcal{G}} |\mathbb{E}_p[g] - \mathbb{E}_q[g]|$. *If* $\theta^* \in \arg\min_\theta \mathcal{L}(\theta)$ *and* $\hat{\theta} \in \arg\min_\theta \hat{\mathcal{L}}_{S,w}(\theta)$, *then*

$$\mathcal{L}(\hat{\theta}) - \mathcal{L}(\theta^*) \leq 2 D_\mathcal{G}(p, q_{S,w}), \quad (2)$$

*where* $D_\mathcal{G}$ *is an instance of an integral probability metric (IPM) (Müller, 1997).*

Given labels are imbalanced, we further decompose the bound class-wise. Assume the target distribution admits a label mixture $p = \sum_y \pi_y^{\mathrm{tar}} p_y$, where $p_y$ is the class-conditional distribution and $\pi^{\mathrm{tar}}$ is the *target* class prior (e.g., the evaluation prior $\pi_y^{\mathrm{uni}}$ or $\pi_y^{\mathrm{emp}}$). Partition $S = \cup_y S_y$ with $|S_y| = m_y$ and write

$$q_{S,w} = \sum_y \rho_y q_y, \qquad \rho_y := \sum_{i \in S_y} w_i,$$
$$q_y := \rho_y^{-1} \sum_{i \in S_y} w_i \delta_{x_i} \ \ (\rho_y > 0). \quad (3)$$

Here $\rho$ is the class distribution *induced* by the weighted pruned objective. Under mild regularity conditions (e.g.,

bounded loss $\|\ell_\theta\|_\infty \le L_{\max}$), for any fixed $\theta$,

$$
\left|\mathbb{E}_p[\ell_\theta] - \mathbb{E}_{q_{S,w}}[\ell_\theta]\right| \le \underbrace{\sum_y \pi_y^{\text{tar}}\left|\mathbb{E}_{p_y}[\ell_\theta] - \mathbb{E}_{q_y}[\ell_\theta]\right|}_{\text{(A) representation error}} \quad (4)
$$
$$
+ \underbrace{2L_{\max}\left\|\pi^{\text{tar}} - \rho\right\|_{\text{TV}}}_{\text{(B) prior mismatch bias}}.
$$

Term (A) measures the approximation quality within each class, while term (B) depends only on how far the induced prior $\rho$ is from the target prior $\pi^{\text{tar}}$. We next analyze the two terms. For term (A), many selection rules yield a per-class approximation rate of the following form.

**Lemma 3.2** (Per-class approximation rate (informal))**.** *With high probability (given $m_y$),*

$$
\sup_{\theta \in \Theta}\left|\mathbb{E}_{p_y}[\ell_\theta] - \mathbb{E}_{q_y}[\ell_\theta]\right| \lesssim \frac{c_y}{m_y^\gamma} + \text{BiasTerm}_y, \quad (5)
$$

*where $c_y$ captures the data/model complexity of class $y$ (we refer to it as* class complexity*). The exponent $\gamma$ depends on the selection strategy; uniform random sampling typically gives $\gamma = \frac{1}{2}$.*

The class complexity $c_y$ is determined by the data and model scale. For norm-controlled neural networks, it scales with the class-conditional input radius and the network norm factors; one standard Rademacher instantiation gives $c_y$ proportional to $R_y \prod_{\ell=1}^L M_\ell$, where $R_y$ bounds the norm of class-$y$ inputs and $M_\ell$ controls the $\ell$-th layer norm. Thus, larger or geometrically more complex classes naturally require more selected samples. The exponent $\gamma$ captures the convergence rate of the selector: i.i.d. sampling corresponds to $\gamma = \frac{1}{2}$, while more structured selection rules can be viewed through the same form with a selector-dependent rate. Appendix A.3 gives the formal bound.

**Theorem 3.3** (Optimal allocation for term (A))**.** *Assume the rate exponent $\gamma$ is the same across classes and ignore rounding effects. The allocation that minimizes $\sum_y \pi_y^{\text{tar}}\left|\mathbb{E}_{p_y}[\ell_\theta] - \mathbb{E}_{q_y}[\ell_\theta]\right|$ under a fixed budget $\sum_y m_y = m$ satisfies*

$$
m_y \propto (c_y \pi_y^{\text{tar}})^k, \qquad k = \frac{1}{1+\gamma}. \quad (6)
$$

This shows that the representation-optimal subset is generally not balanced. Classes with larger $c_y$ (harder classes) and/or larger $\pi_y^{\text{tar}}$ receive more budget. This dependence on $\pi^{\text{tar}}$ is important: different choices of the target prior emphasize different allocations.

For term (B), class-wise reweighting in post-pruning training can resolve the induced prior mismatch. One choice is $w_i = \pi_y^{\text{tar}}/m_y$ for $i \in S_y$, which ensures $\rho_y = \pi_y^{\text{tar}}$ for

all $y$ and eliminates term (B) in (4). The remaining bound depends only on representation:

$$
\text{Error}(\pi^{\text{tar}}) \lesssim \sum_y \pi_y^{\text{tar}}\left(\frac{c_y}{m_y^\gamma} + \text{BiasTerm}_y\right). \quad (7)
$$

The optimal allocation of $\{m_y\}$ follows Equ. (3.3).

**Dilemma.** The analysis above applies to any target prior $\pi^{\text{tar}}$. In this work, we focus on a uniform prior $\pi^{\text{uni}}$ and an empirical prior $\pi^{\text{emp}}$. Our initial analysis shows that a subset optimized for $\pi^{\text{uni}}$ will, in general, differ from one optimized for $\pi^{\text{emp}}$. In short, the optimal condition derived from (A) and (B) reflects an inherent conflict between the two priors across selection and the post-pruning training.

## 4. Method

**Roadmap.** To tackle the dilemma in Sec 3, we first establish base principles that are robust and beneficial across priors, then apply a steering wrapper, as shown in Fig 2. Guided by our theoretical analysis, the method unfolds in three parts: first, we target the prior mismatch bias (Term B) by decoupling structural likelihood from class priors during post-training. Second, we minimize the representation error (Term A) by identifying geometric pruning signals robust to imbalance with a minimum floor. Finally, we introduce a steering wrapper that explicitly modulates the selection strategy to satisfy different user preferences.

### 4.1. Resolving Term B: Robust Post-pruning Distillation

**Decomposing the Dilemma** Hard-label supervision conflates two qualitatively different factors, i.e., the class-conditional structure and the class prevalence. Following Bayes' rule:

$$
\underbrace{\log p(y\mid x)}_{\text{Posterior}} = \underbrace{\log p(x\mid y)}_{\text{Structural Likelihood}} + \underbrace{\log p(y)}_{\text{Class Prior}} + C(x), \quad (8)
$$

where $C(x)$ is independent of $y$. Reweighting simply scales the posterior, implicitly forcing the model to fit a distorted prior. Crucially, in this paradigm, the pruning weights $w$ *define* the target distribution; thus, any slight misalignment in $w$ shifts the global optimum. However, we observe that $\log p(x\mid y)$ describes the semantic geometry of the data manifold, which is *shared* regardless of the evaluation metric (OA or mAcc). The trade-off is fundamentally a conflict of priors, not structures.

**Decoupling via Knowledge Distillation.** To explicitly learn the shared structure separate from the prior offset, we employ Knowledge Distillation (KD). While soft labels are standard in dataset distillation for condensing information density (Yin et al., 2023; Cui et al., 2023), we exploit a distinct, typically overlooked property: KD's *structural invariance* to reweighting. Consider the KD objective on a

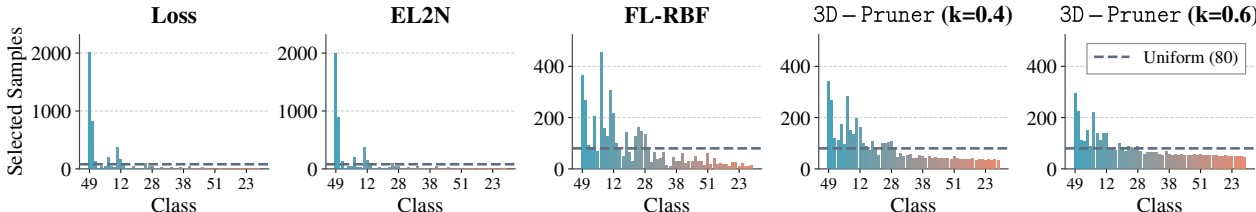

*Figure 3.* Selection composition across algorithms. Incomparable scalar scores (loss, EL2N) bias selection toward many-shot classes. Geometric embedding selection is more robust but still partially affected by class distribution, under-selecting certain classes. SGS mitigates this, also enabling a flexible balancing.

pruned subset $S$ with weights $w$:

$$\hat{\mathcal{L}}_{\mathrm{KD}}^{S,w}(\theta) = \sum_{i \in S} w_i \, \mathrm{CE}\left(T(\cdot \mid x_i), \, f_\theta(\cdot \mid x_i)\right). \quad (9)$$

Crucially, because the teacher's posterior $T(\cdot|x_i)$ is fixed instance-wise, the weights $w$ only modulate the *optimization pace* of each sample, but never alter the *optimal target* itself. This stands in sharp contrast to hard-label supervision, where $w$ constitutes the target prior. KD renders the learning objective theoretically robust to the reweighting. This distinction also appears at the optimization level. If a pruned subset induces prior $\rho$ but hard-label training targets $\pi^{\mathrm{tar}}$, correcting the mismatch requires class weights $a_y = \pi_y^{\mathrm{tar}}/\rho_y$. With per-sample gradient norms bounded by $G$, the standard second-moment bound scales as

$$\mathbb{E}\|\hat{g}\|^2 \leq G^2 \sum_y \frac{(\pi_y^{\mathrm{tar}})^2}{\rho_y}, \qquad \sum_y \frac{(\pi_y^{\mathrm{tar}})^2}{\rho_y} \geq 1, \quad (10)$$

with equality in the second inequality only when $\rho = \pi^{\mathrm{tar}}$. KD absorbs this correction into instance-wise teacher targets, allowing the student to train on the pruned subset with instance-balanced sampling.

**Proposition 4.1** (Weight-robustness of KD)**.** *Assume the student is expressive enough to interpolate the teacher on the support $S$, namely $f_\theta(\cdot \mid x_i) = T(\cdot \mid x_i)$ for all $i \in S$. Then for any strictly positive weights $w$, any interpolating solution is a global minimizer of $\hat{\mathcal{L}}_{\mathrm{KD}}^{S,w}$.*

*Insight.* KD effectively shifts the bottleneck from tuning the weights $w$ to calibrating the teacher $T$. By fixing the reference for the structural likelihood, the student preserves the correct manifold geometry regardless of the re-balancing.

**Optimizing the Structure.** In 3D recognition, standard protocols typically minimize average error without rebalancing. Consequently, a teacher trained under this regime inevitably encodes the long-tailed prior, producing biased targets that conflate high-quality structural signals with class prevalence. To extract a refined *structural likelihood*, we employ a twofold refinement targeting both boundary alignment and manifold geometry.

*1) Calibrating the Structural Source (Boundary Alignment).* First, we mitigate the prior bias in the teacher's logits to

*Table 2.* **KD improves accuracy and reduces sensitivity to rebalancing.** Rebalancing policies include instance-balanced sampling (IB), class-balanced sampling (CB), and square-root sampling (Sqrt) (Kang et al., 2019), plus class-balanced loss (CB-Loss) (Cui et al., 2019). Results are obtained on ShapeNet55 with PointNet++, with imbalanced set selected by FL-RBF (Wei et al., 2015).

| | Hard | | KD | |
|---|---|---|---|---|
| Rebalancing | OA | mAcc | OA | mAcc |
| CB-Samp. (CB) | 79.02 | 61.20 | 82.09 | 64.56 |
| CB-Loss | 77.45 | 62.16 | 81.94 | 63.57 |
| IB-Samp. (IB) | 79.92 | 58.41 | 82.04 | 64.02 |
| Sqrt-Samp. (Sqrt) | 80.00 | 60.10 | 81.21 | 64.81 |
| **Mean±Std** | **79.10±1.18** | **60.47±1.61** | **81.82±0.41** | **64.24±0.56** |

correct the sample-to-boundary relations. Recognizing that the learned embedding $\phi$ already captures robust semantic geometry (Kang et al., 2019), we freeze $\phi$ and re-train only the classifier head $(W, b)$ using a class-balanced objective on the full dataset:

$$\min_{W,b} \mathbb{E}_{p_{\mathrm{train}}}\left[\alpha_y \, \mathrm{CE}\left(\delta_y, \sigma(W\phi(x) + b)\right)\right]. \quad (11)$$

This adjustment neutralizes the head's tendency to favor majority classes, aligning the logits closer to the intrinsic likelihood $p(x|y)$.

*2) Preserving Geometry under Sparsity (Intrinsic Topology).* While calibration refines the decision boundaries, pruning introduces a density problem: the learned embedding manifold collapses into sparse anchor points. Here, geometry means the topology of the teacher's representation space. Structural information includes the *extrinsic* distance to decision boundaries (via logits) and the *intrinsic* relational topology among samples; we explicitly transfer the latter using Relational Knowledge Distillation (RKD) (Park et al., 2019). RKD enforces consistency in pairwise distances ($\psi_D$) and triplet angles ($\psi_A$) within a batch $\mathcal{B}$:

$$\mathcal{L}_{\mathrm{RKD}} = \sum_{\mathcal{B}^2} \ell_\delta(\psi_D^T, \psi_D^S) + \sum_{\mathcal{B}^3} \ell_\delta(\psi_A^T, \psi_A^S). \quad (12)$$

By anchoring pairwise distances and triplet angles in the teacher embedding, RKD enables the student to reconstruct the *internal shape* of the class manifold even from sparse samples. We use the standard RKD formulation; the 3D specificity comes from applying it to geometry-aware point-cloud representations produced by the teacher.

## 4.2. Resolving Term A: Geometry-aware Selection

**1) Robust Signals: Leveraging 3D Inductive Bias.** The first challenge in selection is identifying a pruning metric that remains comparable across highly imbalanced classes. We believe the importance of signal quality outweighs that of further allocation based on the signal. We identify *embedding geometry* as the robust signal for 3D data, distinct from classifier-derived scalars (e.g., Loss, Entropy, EL2N).

In 3D recognition, models rely on deep inductive biases to learn local geometric primitives (e.g., corners, flats, curvatures). Crucially, these geometric units are *shared* across both few-shot and many-shot classes, making the embedding space inherently more stable and comparable than the classifier decision boundaries. In contrast, scalar signals are entangled with the training prior: as shown in Table 3, metrics like Loss and EL2N exhibit extreme dependence on class frequency, collapsing selection onto head classes (as shown in Fig 3). Furthermore, selecting based on embedding geometry aligns with our post-pruning distillation objective, which explicitly preserves this structure via RKD.

*Table 3.* **Signal Audit.** We measure correlation ($\rho$) and dependence ($R^2$) of class-aggregated signal magnitude on class size, and head/tail score overlap. For embeddings, magnitude is distance to the class center. **Embedding signals show minimal class-frequency dependence and high cross-class overlap**, whereas scalar signals (Loss, EL2N) are dominated by the training prior, causing extreme selection imbalance (up to $40\times$).

| Signal | $\rho$ (Corr. w/ Size) $\downarrow$ | $R^2$ (Dep. on Size) $\downarrow$ | Overlap $\uparrow$ | Imbalance Ratio $\downarrow$ |
|---|---|---|---|---|
| **Embedding** | **-0.306** | **0.148** | **96%** | **1.88x** |
| Loss | 0.635 | 0.145 | 26% | 40.94x |
| EL2N | 0.594 | 0.164 | 30% | 33.01x |
| Entropy | 0.731 | 0.481 | 26% | 7.84x |

**2) A Prior-Agnostic High-Return Regime.** Even with a robust signal, pure global selection naturally drifts towards majority classes due to their density. From Theorem 3.3, we know a single fixed subset cannot be simultaneously optimal for divergent target priors (e.g., OA vs. mAcc). However, the error bound reveals a *shared high-return regime* that exists before any target-specific trade-off kicks in.

Let $E_y(m_y) \propto m_y^{-\gamma}$ denote the power-law error decay. The marginal gain from the first few samples is extremely steep. Specifically, allocating a minimal floor of $b$ samples captures a constant fraction of the reducible error:

$$\frac{E_y(1) - E_y(b)}{E_y(1)} = 1 - \frac{1}{b^\gamma}. \tag{13}$$

Crucially, this relative gain across all classes is *independent* of any reweighting induced by the target prior $\pi^{\text{tar}}$. This motivates our strategy to reserve a safety budget to guarantee a minimum floor $m_y \geq b$ for every class. By securing this "safety floor," we harvest the common high-curvature region of the error surface, preventing systematic under-coverage of few-shot classes regardless of the final evaluation metric.

**Seeded Global Selection (SGS): Steering the Prior.** Finally, we unify the proposed principles into a streamlined selection wrapper. Having established that a robust subset requires both a *safety floor* (to capture the shared high-return regime) and *geometry-aware selection* (to leverage robust signals), SGS is designed as a direct instantiation of these two requirements. It interpolates between a guaranteed floor and a data-driven distribution using a single steering parameter $K \in [0, 1]$.

SGS directly instantiates the Term-(A) bound. Let

$$\mathcal{B}_A(S; \pi^{\text{tar}}) := \sum_y \pi_y^{\text{tar}} \left( \frac{c_y}{m_y^\gamma} + \text{BiasTerm}_y \right)$$

denote the residual Term-(A) upper bound after removing the prior-mismatch term. Suppose SGS selects $s_y$ seed samples and $g_y$ global-stage samples from class $y$, so that $m_y = s_y + g_y$. Then

$$\mathcal{B}_A^{\text{SGS}}(\pi^{\text{tar}}) = \underbrace{\sum_y \pi_y^{\text{tar}} \left( \frac{c_y}{s_y^\gamma} + \text{BiasTerm}_y \right)}_{\text{seed baseline}} \\ - \underbrace{\sum_y \pi_y^{\text{tar}} c_y \left( s_y^{-\gamma} - (s_y + g_y)^{-\gamma} \right)}_{\text{global residual gain}}. \tag{14}$$

The seed stage provides a prior-robust floor: if $s_y \geq b$ for every class, the baseline is uniformly bounded by

$$\sum_y \pi_y^{\text{tar}} \left( \frac{c_y}{b^\gamma} + \text{BiasTerm}_y \right).$$

The global stage then gives a nonnegative residual gain, strictly positive for any class receiving additional global samples. Adjusting $K$ changes how much budget is used for the seed floor versus global refinement, making $K$ a direct control knob over the OA–mAcc preference. Operationally, SGS wraps a base selector $\phi$: it first allocates $b = \lfloor KB/|\mathcal{C}| \rfloor$ samples to each class, then fills the remaining budget with global selection. Algorithm 1 details the procedure.

## 5. Experiment

### 5.1. Experiment Setup

**Datasets.** We evaluate on point clouds (ModelNet40 (Wu et al., 2015), ScanObjectNN (Uy et al., 2019), ShapeNet55 (Yi et al., 2016)) and meshes (ModelNet40). **Models.** For point clouds, we use CNN/MLP (PointNeXt (Qian et al., 2022), PointVector (Deng et al., 2023), PointNet++ (Qi et al., 2017)) and transformer-based PointMAE (Pang et al., 2023). For meshes, we use MeshMAE (Liang et al., 2022) and MeshNet (Feng et al., 2019).

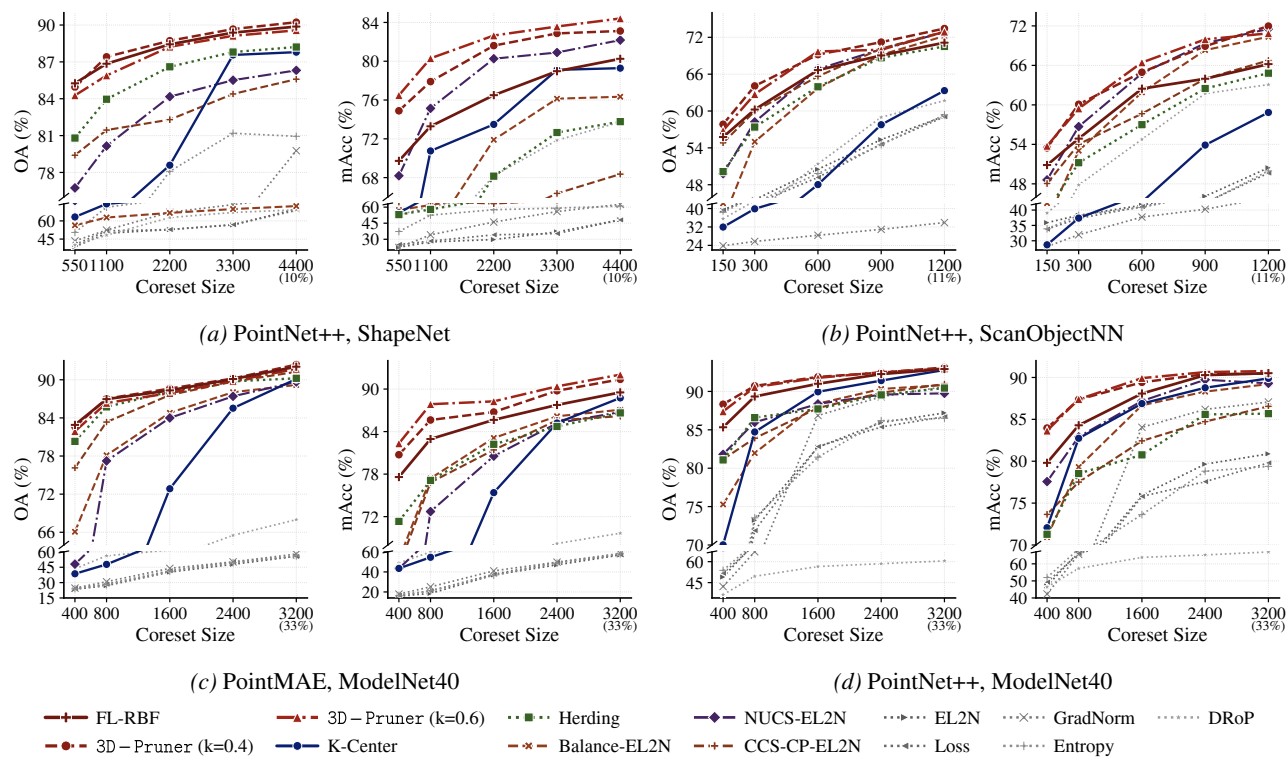

*Figure 4.* Comparison of pruning methods across various datasets, models, and budgets on point cloud modality.

Legend:
— FL-RBF, — 3D−Pruner (k=0.6), ⋯ Herding, — NUCS-EL2N, ⋯ EL2N, ⋯ GradNorm, ⋯ DRoP
— 3D−Pruner (k=0.4), — K-Center, — Balance-EL2N, — CCS-CP-EL2N, ⋯ Loss, ⋯ Entropy

**Algorithm 1** Seeded Global Selection (SGS)

**Input** : Dataset $\mathcal{D}$, Budget $B$, Safety ratio $K$, Classes $\mathcal{C}$, Selector $\phi(\mathcal{X}, n, \mathcal{S}_{init})$ selecting $n$ items from $\mathcal{X}$ given seed $\mathcal{S}_{init}$

**Output** : Final Coreset $\mathcal{S}$

1  $b \leftarrow \lfloor (K \cdot B)/|\mathcal{C}| \rfloor$

2  $\mathcal{S}_{\text{strat}} \leftarrow \bigcup_{c \in \mathcal{C}} \phi(\mathcal{D}_c, b, \emptyset), \quad \mathcal{S}_{\text{glob}} \leftarrow \phi(\mathcal{D}, B - b|\mathcal{C}|, \mathcal{S}_{\text{strat}})$

3  $\mathcal{S}_{\text{union}} \leftarrow \mathcal{S}_{\text{strat}} \cup \mathcal{S}_{\text{glob}}$

4  **if** $|\mathcal{S}_{\text{union}}| < B$ **then**

5  $\quad \lfloor \quad \mathcal{S}_{\text{union}} \leftarrow \phi(\mathcal{D}, B - |\mathcal{S}_{\text{union}}|, \mathcal{S}_{\text{union}})$

6  **return** $\mathcal{S}_{\text{union}}$

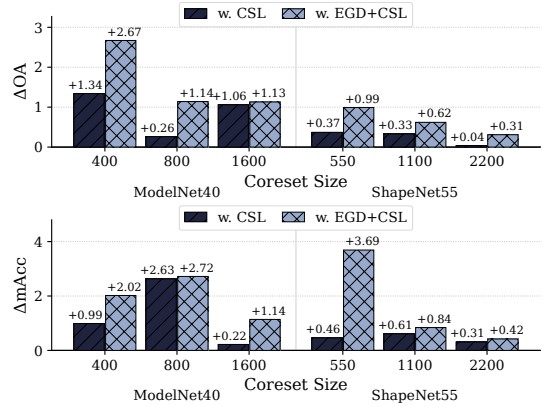

*Figure 5.* Impact of robust distillation methods: Calibrated Soft Label (CSL) and Embedding Geometry Distillation (EGD) on OA and mAcc. Results are obtained on PointNet++.

**Methods.** We evaluate classical pruning metrics spanning scalar difficulty signals and embedding-geometric selectors: loss, gradient norm, EL2N (Paul et al., 2021b), entropy (Wang & Shang, 2014), herding (Welling, 2009), K-center Greedy (Sener & Savarese, 2017b), and FL-RBF (Wei et al., 2015). We also compare imbalance-aware methods including DRoP (Vysogorets et al., 2024), NUCS (Zhang et al., 2025), and CCS-CP (Tsai et al., 2025).

**Metrics & Budget.** Following Sec. 2, we report OA and mAcc per standard protocols. We focus on the high-compression regime to *explicitly* distinguish dataset pruning from head-class downsampling (Buda et al., 2018). Given the extreme skew (e.g., in ShapeNet55: 84.9% Many-shot

vs. 1.6% Few-shot), this setting isolates *pure* selection efficacy under strict constraints (extended results on relaxed budgets, additional results on different models, and detailed experimental settings provided in the appendix).

### 5.2. Main Results

Fig. 4 demonstrates the superiority of our 3D-Pruner (based on FL-RBF) compared to other pruning methods. Across the evaluated datasets, models, and budgets, 3D-Pruner achieves the strongest overall profile across OA and mAcc: it substantially improves mAcc while maintaining strong OA, and different choices of $K$ provide oper-

*Table 4.* ***Fair comparison*** of selection strategies on *ModelNet40*, *ScanObjectNN*, and *ShapeNet55* with Calibrated Soft Label and Embedding Geometry Distillation from `3D-Pruner` applied to all baselines.

| | | ModelNet40 | | | | | | ScanObjectNN | | | | | | ShapeNet55 | | | | | |
| | | m = 400 | | m = 800 | | m = 1600 | | m = 150 | | m = 300 | | m = 600 | | m = 550 | | m = 1100 | | m = 2200 | |
| Metric Type | Selection Policy | OA | mAcc | OA | mAcc | OA | mAcc | OA | mAcc | OA | mAcc | OA | mAcc | OA | mAcc | OA | mAcc | OA | mAcc |
|---|---|---|---|---|---|---|---|---|---|---|---|---|---|---|---|---|---|---|---|
| **Scalar Scores** | *Global Selection* | | | | | | | | | | | | | | | | | | |
| | Loss | 56.36 | 47.76 | 75.97 | 65.97 | 88.24 | 82.43 | 35.84 | 35.60 | 52.36 | 51.37 | 63.15 | 61.10 | 55.21 | 20.12 | 60.13 | 25.34 | 67.87 | 25.38 |
| | GradNorm | 44.61 | 35.72 | 70.22 | 59.32 | 87.96 | 79.62 | 45.70 | 36.22 | 51.84 | 42.47 | 60.40 | 50.63 | 62.82 | 22.60 | 67.39 | 27.18 | 71.06 | 31.54 |
| | Entropy | 58.71 | 49.24 | 77.84 | 68.37 | 88.37 | 82.74 | 34.46 | 34.80 | 50.97 | 49.68 | 63.21 | 62.30 | 66.47 | 24.06 | 75.59 | 33.00 | 79.57 | 41.27 |
| | EL2N | 55.27 | 44.80 | 73.54 | 64.81 | 88.41 | 82.83 | 35.77 | 35.61 | 51.39 | 50.62 | 63.98 | 62.90 | 52.23 | 19.45 | 62.53 | 23.17 | 65.81 | 24.97 |
| | *Stratified Sampling* | | | | | | | | | | | | | | | | | | |
| | DRoP | 41.86 | 45.02 | 55.63 | 57.47 | 64.82 | 65.32 | 38.06 | 39.77 | 46.29 | 48.54 | 58.39 | 58.71 | 45.50 | 54.73 | 54.79 | 60.72 | 64.45 | 69.40 |
| | Balance-EL2N | 74.39 | 69.39 | 83.23 | 81.43 | 89.82 | 88.70 | 49.34 | 49.14 | 59.02 | 56.92 | 65.68 | 64.02 | 78.53 | 74.45 | 82.89 | 79.26 | 84.46 | 81.79 |
| | NUCS-EL2N | 82.41 | 78.72 | 86.26 | 85.04 | 89.62 | 88.51 | 54.41 | 52.46 | 61.76 | 59.82 | 67.93 | 66.77 | 78.17 | 71.89 | 80.66 | 75.90 | 85.02 | 79.92 |
| | CCS-CP-EL2N | 81.44 | 74.55 | 85.49 | 79.71 | 89.34 | 82.39 | 55.59 | 48.89 | 61.69 | 54.87 | 68.25 | 62.12 | 80.90 | 70.30 | 83.81 | 66.43 | 84.81 | 67.80 |
| **Vector Emb.** | *Global Selection* | | | | | | | | | | | | | | | | | | |
| | Herding | 81.89 | 74.25 | 87.70 | 80.68 | 89.10 | 83.84 | 50.21 | 44.63 | 59.54 | 53.77 | 66.58 | 61.11 | 82.21 | 60.02 | 83.99 | 62.18 | 86.98 | 70.97 |
| | K-Center | 75.61 | 75.82 | 86.51 | 84.29 | 90.80 | 88.08 | 36.16 | 32.25 | 42.05 | 39.09 | 54.19 | 53.89 | 68.91 | 64.72 | 80.99 | 72.73 | 80.14 | 75.07 |
| | FL-RBF | 88.00 | 81.83 | 90.48 | 87.01 | 92.13 | 89.22 | 55.56 | 50.13 | 62.11 | 56.99 | 67.41 | 63.09 | 85.52 | 68.36 | 87.71 | 73.25 | 88.77 | 78.32 |
| | **SGS (Ours)** | | | | | | | | | | | | | | | | | | |
| | FL-RBF (K=0.4) | 88.33 | 83.96 | 90.72 | 87.37 | 91.89 | 89.42 | 57.84 | 53.47 | 64.08 | 60.09 | 69.18 | 64.96 | 84.95 | 74.89 | 87.41 | 77.90 | 88.72 | 81.61 |
| | Δ vs FL-RBF | +0.33 | +2.13 | +0.24 | +0.36 | -0.24 | +0.20 | +2.28 | +3.34 | +1.97 | +3.10 | +1.77 | +1.87 | -0.57 | +6.53 | -0.30 | +4.65 | -0.05 | +3.29 |
| | FL-RBF (K=0.6) | 87.39 | 83.64 | 90.56 | 87.40 | 91.77 | 89.92 | 56.83 | 53.76 | 62.73 | 59.44 | 69.70 | 66.44 | 84.27 | 76.49 | 85.89 | 80.30 | 88.23 | 82.65 |
| | Δ vs FL-RBF | -0.61 | +1.81 | +0.08 | +0.39 | -0.36 | +0.70 | +1.27 | +3.63 | +0.62 | +2.45 | +2.29 | +3.35 | -1.25 | +8.13 | -1.82 | +7.05 | -0.54 | +4.33 |

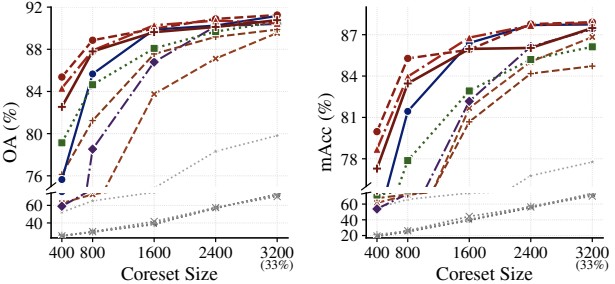

*Figure 6.* Comparison of pruning methods on mesh modality, legend following Fig. 4

ating points along the OA–mAcc preference curve. We next separately analyze their effects in detail.

***Effectiveness of Term (B) Principles.*** Term (B) principles are realized through Calibrated Soft Labels (CSL) combined with Embedding Geometry Distillation (EGD). Fig. 5 demonstrates the gradual improvement achieved by these two methods. Improvements in both mAcc and OA can be observed after adding CSL, with further gains when incorporating EGD. This confirms our analysis that an improved structural likelihood can benefit across different priors. There is also a generally decaying trend when both EGD and CSL are applied under increasing budgets. This is because EGD primarily helps to mitigate the data scarcity issue by transferring information about the internal structure of the teacher's embedding; as more samples are selected, its effect weakens.

***Effectiveness of Term (A) Principles.*** First, we isolate the impact of the selection signal. By applying our EGD and CSL framework across all baselines upon PointNet++ (Tab. 4), we find that naive global selection using **embedding signal** consistently outperforms classifier-derived scalar signals, confirming the audit in Tab. 3. These scalar signals are less class-comparable, therefore the selection is essential broken. Crucially, since scalar signals correlate heavily with class size, methods that rely on them to estimate

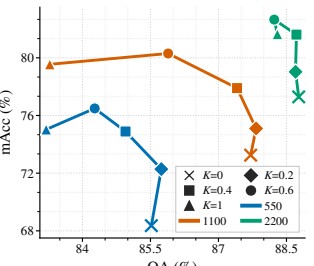

"difficulty" (e.g., DRoP, NUCS, CCS-CP) can improve results yet still unreliable under such setting. On ShapeNet55, these complex allocators are outperformed even by simple stratified sampling. This validates our first principle: in the presence of strong inductive

*Figure 7.* Preference steering with different $K$.

bias but weak scalar calibration, *what you measure* (embedding geometry) matters more than *how you allocate* based on noisy metrics. Second, we analyze the necessity of the safety floor by varying the steering parameter $K$ in SGS (Fig. 7). The trajectory reveals two distinct regimes, validating our theory.

**(i) Shared High-Reward Regime ($K \leq 0.2$):** Increasing the floor from $K = 0$ (pure global) significantly boosts mAcc with negligible or no cost to OA. This confirms our second principle: a minimum floor is beneficial across different target priors.

**(ii) Trade-off Regime ($K > 0.2$):** Beyond the safety floor, a Pareto front emerges, allowing users to exchange OA for mAcc according to preference. Notably, the pure mechanical balance ($K = 1$) is often Pareto-dominated, reflecting its inability to adapt to the varying class complexities $c_y$ inherent in Theorem 3.3 (optimal allocation). We additionally repeat representative ShapeNet55 experiments over three seeds (42, 0, 1), focusing on the high-compression regime where pruning choices matter most. Table 5 shows that the main pattern is stable: CSL and EGD strengthen the base selector, while SGS controls the OA–mAcc preference

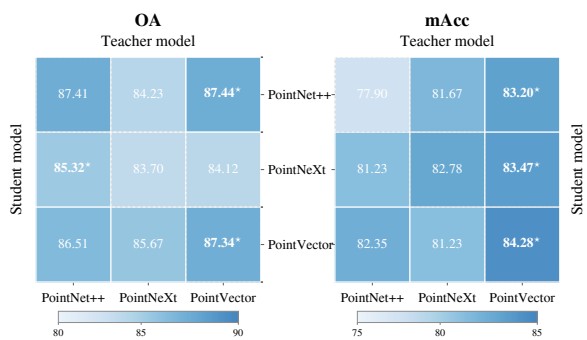

*Figure 8.* Transferability of `3D-Pruner` across different architectures. Results are based on $m = 1100$ on ShapeNet55.

through $K$. Smaller $K$ gives the strongest OA, and larger $K$ shifts the subset toward stronger mAcc. Appendix C.1 reports the full three-budget version.

*Table 5.* Three-seed ShapeNet55 evaluation with PointNet++. Each entry is OA/mAcc.

| Setting | $m = 550$ | $m = 2200$ |
|---|---|---|
| FL-RBF | 83.4±0.4/65.2±0.8 | 87.2±0.2/74.2±0.1 |
| + CSL + EGD | 84.6±0.4/68.4±0.7 | 88.7±0.2/78.3±0.3 |
| + SGS ($K = 0.2$) | **85.2±0.1**/72.3±0.3 | **88.9±0.2**/79.0±0.2 |
| + SGS ($K = 0.4$) | 84.5±0.0/**73.8±0.3** | 88.5±0.2/**80.7±0.4** |

### 5.3. Additional Results

**Cross-Architecture Transfer.** We further validate whether our shared principles hold under cross-architecture pruning, where the teacher and student models have different architectures. Since EGD focuses solely on likelihood structure, we adopt RKD, which is free from embedding dimension constraints. Results in Fig. 8 demonstrate that cross-architecture pruning and training can be achieved with limited data (only 1,100 images). Notably, the student's performance is not hindered by architectural differences and can even benefit from a stronger teacher. For instance, PointNet++ and PointNext show improved mAcc when using PointVector as the teacher, highlighting the effectiveness of our principles in a broader setting.

**Generalization to Other Modalities.** Finally, we extend beyond the point cloud modality and show that our strategy can transfer to other modalities. Fig. 6 shows the results on the MeshNet with `3D-Pruner` leading both OA and mAcc, validating that the effectiveness of our shared principles is not limited to point cloud, showing the potential to extend to other modalities as well.

## 6. Conclusion

In this work, we explored 3D dataset pruning under inherent class imbalance, a setting where the divergence between overall and mean accuracy often imposes difficult design choices. Rather than accepting a premature trade-off, we

leveraged a quadrature error decomposition to identify a shared optimization path beneficial to both priors. This theoretical insight is realized in our `3D-Pruner` framework: we first secure a high performance floor by targeting the universally beneficial components—reducing prior-mismatch via calibrated geometric distillation and minimizing representation error through safety-aware selection. Finally, we manage the irreducible preference divergence through a lightweight steering wrapper. Experiments confirm that this "principled foundation first, steering second" strategy establishes superior performance floors across diverse architectures and evaluation priors.

## Acknowledgements

This work is supported by the MBZUAI-WIS Joint Program for Artificial Intelligence Research.

## Impact Statement

This paper presents work whose goal is to advance the field of Machine Learning. There are many potential societal consequences of our work, none of which we feel must be specifically highlighted here.

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

# Appendix

**Roadmap.** This appendix provides supporting materials organized as follows:

- **Section A**: Mathematical proofs for the generalization gap, class-wise error decomposition, Rademacher complexity bounds, optimal budget allocation, the SGS Term-(A) bound, and distillation robustness.

- **Section B**: Detailed experimental settings, including model architectures for PointNeXt and MeshMAE, and the three-phase training protocols.

- **Section C**: Extended empirical evaluations, including multi-seed ShapeNet55 results, diverse 3D architectures, and performance analysis under relaxed data pruning budgets.

## A. Proof

### A.1. Proof for Lemma 3.1

*Proof.* Let $q = q_{S,w}$ for brevity and $D := D_{\mathcal{G}}(p, q)$. For any $\theta$,

$$\mathcal{L}(\theta) \leq \hat{\mathcal{L}}_q(\theta) + D \quad \text{and} \quad \hat{\mathcal{L}}_q(\theta) \leq \mathcal{L}(\theta) + D$$

by the definition of $D$ (taking $g = \ell_\theta$). Since $\hat{\theta}$ minimizes $\hat{\mathcal{L}}_q$,

$$\hat{\mathcal{L}}_q(\hat{\theta}) \leq \hat{\mathcal{L}}_q(\theta^*).$$

Therefore,

$$\mathcal{L}(\hat{\theta}) \leq \hat{\mathcal{L}}_q(\hat{\theta}) + D \leq \hat{\mathcal{L}}_q(\theta^*) + D \leq \mathcal{L}(\theta^*) + 2D,$$

which proves the claim. $\qquad\square$

### A.2. Proof for Class-wise Decomposition

*Proof.* Expand:

$$\mathbb{E}_p[\ell_\theta] - \mathbb{E}_q[\ell_\theta] = \sum_y \pi_y^{\text{tar}} \mathbb{E}_{p_y}[\ell_\theta] - \sum_y \rho_y \mathbb{E}_{q_y}[\ell_\theta]$$

$$= \sum_y \pi_y^{\text{tar}} \left( \mathbb{E}_{p_y}[\ell_\theta] - \mathbb{E}_{q_y}[\ell_\theta] \right) + \sum_y (\pi_y^{\text{tar}} - \rho_y) \mathbb{E}_{q_y}[\ell_\theta].$$

Take absolute values and apply triangle inequality:

$$\leq \sum_y \pi_y^{\text{tar}} \left| \mathbb{E}_{p_y}[\ell_\theta] - \mathbb{E}_{q_y}[\ell_\theta] \right| + \sum_y \left| \pi_y^{\text{tar}} - \rho_y \right| \cdot \left| \mathbb{E}_{q_y}[\ell_\theta] \right|.$$

Since $|\ell_\theta| \leq L_{\max}$, we have $|\mathbb{E}_{q_y}[\ell_\theta]| \leq L_{\max}$. Thus

$$\leq \sum_y \pi_y^{\text{tar}} \left| \mathbb{E}_{p_y}[\ell_\theta] - \mathbb{E}_{q_y}[\ell_\theta] \right| + L_{\max} \sum_y \left| \pi_y^{\text{tar}} - \rho_y \right| = \sum_y \pi_y^{\text{tar}} \cdot |\cdot| + 2L_{\max} \|\pi^{\text{tar}} - \rho\|_{\text{TV}}.$$

The uniform version follows by taking $\sup_\theta$ on both sides. $\qquad\square$

### A.3. Proof for Lemma 3.2

**Lemma A.1** (Class-wise error bound (formal)). *Fix a class $y$. Let $S_y = \{x_1^{(y)}, \ldots, x_{m_y}^{(y)}\}$ be drawn i.i.d. from the class-conditional distribution $p_y$, and define the empirical measure*

$$q_y := \frac{1}{m_y} \sum_{i=1}^{m_y} \delta_{x_i^{(y)}}.$$

*Assume the loss is bounded: $0 \leq \ell_\theta(x) \leq L_{\max}$ for all $\theta \in \Theta$ and $x$. Let $\mathcal{G}_y := \{\ell_\theta(\cdot) : \theta \in \Theta\}$ and define the empirical Rademacher complexity*

$$\widehat{\mathfrak{R}}_{S_y}(\mathcal{G}_y) := \mathbb{E}_\sigma \left[ \sup_{g \in \mathcal{G}_y} \frac{1}{m_y} \sum_{i=1}^{m_y} \sigma_i g\left(x_i^{(y)}\right) \right],$$

*where $\sigma_1, \ldots, \sigma_{m_y}$ are i.i.d. Rademacher variables. Then for any $\delta \in (0, 1)$, with probability at least $1 - \delta$ over $S_y$,*

$$\sup_{\theta \in \Theta} \left| \mathbb{E}_{p_y}[\ell_\theta] - \mathbb{E}_{q_y}[\ell_\theta] \right| \leq 2\widehat{\mathfrak{R}}_{S_y}(\mathcal{G}_y) + 3L_{\max}\sqrt{\frac{\log(4/\delta)}{2m_y}}. \tag{15}$$

*Moreover, if (for the considered hypothesis class) there exists a constant $c_y > 0$ such that*

$$\widehat{\mathfrak{R}}_{S_y}(\mathcal{G}_y) \leq \frac{c_y}{2\sqrt{m_y}}, \tag{16}$$

*then with the same probability,*

$$\sup_{\theta \in \Theta} \left| \mathbb{E}_{p_y}[\ell_\theta] - \mathbb{E}_{q_y}[\ell_\theta] \right| \leq \frac{c_y}{\sqrt{m_y}} + 3L_{\max}\sqrt{\frac{\log(4/\delta)}{2m_y}}. \tag{17}$$

*Proof.* Define the normalized loss class

$$\widetilde{\mathcal{G}}_y := \left\{ x \mapsto \frac{\ell_\theta(x)}{L_{\max}} : \theta \in \Theta \right\}.$$

Since $0 \leq \ell_\theta(x) \leq L_{\max}$, every function in $\widetilde{\mathcal{G}}_y$ maps into $[0, 1]$. Apply the Rademacher-complexity generalization bound (Theorem 3.3 in (Mohri et al., 2018)) to the class $\widetilde{\mathcal{G}}_y$ with confidence parameter $\delta/2$. With probability at least $1 - \delta/2$, the following holds simultaneously for all $\theta \in \Theta$:

$$\mathbb{E}_{p_y}\left[ \frac{\ell_\theta}{L_{\max}} \right] \leq \mathbb{E}_{q_y}\left[ \frac{\ell_\theta}{L_{\max}} \right] + 2\widehat{\mathfrak{R}}_{S_y}(\widetilde{\mathcal{G}}_y) + 3\sqrt{\frac{\log(4/\delta)}{2m_y}}. \tag{18}$$

To obtain the reverse inequality, apply the same theorem to the class $1 - \widetilde{\mathcal{G}}_y := \{x \mapsto 1 - g(x) : g \in \widetilde{\mathcal{G}}_y\}$ (which also maps into $[0, 1]$) with confidence parameter $\delta/2$. With probability at least $1 - \delta/2$, for all $\theta \in \Theta$,

$$\mathbb{E}_{p_y}\left[ 1 - \frac{\ell_\theta}{L_{\max}} \right] \leq \mathbb{E}_{q_y}\left[ 1 - \frac{\ell_\theta}{L_{\max}} \right] + 2\widehat{\mathfrak{R}}_{S_y}(1 - \widetilde{\mathcal{G}}_y) + 3\sqrt{\frac{\log(4/\delta)}{2m_y}}.$$

Rearranging gives

$$\mathbb{E}_{p_y}\left[ \frac{\ell_\theta}{L_{\max}} \right] \geq \mathbb{E}_{q_y}\left[ \frac{\ell_\theta}{L_{\max}} \right] - 2\widehat{\mathfrak{R}}_{S_y}(1 - \widetilde{\mathcal{G}}_y) - 3\sqrt{\frac{\log(4/\delta)}{2m_y}}. \tag{19}$$

By a union bound over the two events, with probability at least $1 - \delta$, both (18) and (19) hold simultaneously for all $\theta$. Noting that $\widehat{\mathfrak{R}}_{S_y}(1 - \widetilde{\mathcal{G}}_y) = \widehat{\mathfrak{R}}_{S_y}(\widetilde{\mathcal{G}}_y)$ (translation by a constant does not change Rademacher complexity, and sign-flips are absorbed by Rademacher symmetry), we obtain

$$\sup_{\theta \in \Theta} \left| \mathbb{E}_{p_y}\left[ \frac{\ell_\theta}{L_{\max}} \right] - \mathbb{E}_{q_y}\left[ \frac{\ell_\theta}{L_{\max}} \right] \right| \leq 2\widehat{\mathfrak{R}}_{S_y}(\widetilde{\mathcal{G}}_y) + 3\sqrt{\frac{\log(4/\delta)}{2m_y}}.$$

Multiplying by $L_{\max}$ yields (15) and using $\widehat{\mathfrak{R}}_{S_y}(\widetilde{\mathcal{G}}_y) = \widehat{\mathfrak{R}}_{S_y}(\mathcal{G}_y)/L_{\max}$ gives the form stated in the lemma.

Finally, condition (16) is satisfied by many hypothesis classes. In particular, for spectrally-controlled neural networks, the scale-sensitive complexity depends on data norm and products of spectral norms (see, e.g., (Bartlett et al., 2017)); absorbing the resulting constants/log factors into $c_y$ yields (16), and plugging it into (15) gives (17). $\square$

## A.4. Proof for Theorem 3.3

**Theorem A.2** (Optimal allocation for Term (A)). *Let $\gamma > 0$ be a common exponent across classes, and let $\pi_y^{\mathrm{tar}} > 0$, $c_y > 0$. Consider the continuous relaxation of the budget allocation problem:*

$$\min_{\{m_y > 0\}} \sum_y \pi_y^{\mathrm{tar}} \frac{c_y}{m_y^{\gamma}} \quad s.t. \quad \sum_y m_y = m,$$

*where $m > 0$ is the total budget. Then the unique optimizer satisfies*

$$m_y^* = m \cdot \frac{(\pi_y^{\mathrm{tar}} c_y)^{\frac{1}{1+\gamma}}}{\sum_{y'} (\pi_{y'}^{\mathrm{tar}} c_{y'})^{\frac{1}{1+\gamma}}} \quad \Longleftrightarrow \quad m_y^* \propto (\pi_y^{\mathrm{tar}} c_y)^{\frac{1}{1+\gamma}}. \tag{20}$$

*Proof.* Define $a_y := \pi_y^{\mathrm{tar}} c_y > 0$ and consider

$$\min_{\{m_y > 0\}} \sum_y a_y m_y^{-\gamma} \quad s.t. \quad \sum_y m_y = m.$$

Form the Lagrangian

$$\mathcal{J}(m_1, \ldots, m_C, \lambda) = \sum_y a_y m_y^{-\gamma} + \lambda \Big( \sum_y m_y - m \Big).$$

Taking derivatives and setting them to zero yields, for each class $y$,

$$\frac{\partial \mathcal{J}}{\partial m_y} = -\gamma a_y m_y^{-(\gamma+1)} + \lambda = 0 \quad \Longrightarrow \quad m_y^{\gamma+1} = \frac{\gamma a_y}{\lambda}.$$

Thus $m_y \propto a_y^{1/(1+\gamma)}$. Enforcing the constraint $\sum_y m_y = m$ gives

$$m_y^* = m \cdot \frac{a_y^{1/(1+\gamma)}}{\sum_{y'} a_{y'}^{1/(1+\gamma)}} = m \cdot \frac{(\pi_y^{\mathrm{tar}} c_y)^{\frac{1}{1+\gamma}}}{\sum_{y'} (\pi_{y'}^{\mathrm{tar}} c_{y'})^{\frac{1}{1+\gamma}}}.$$

This is the stated solution (20). $\qquad\square$

## A.5. SGS Bound for Term (A)

We derive the bound used in the SGS discussion. Recall the residual Term-(A) quantity

$$\mathcal{B}_A(S; \pi^{\mathrm{tar}}) := \sum_y \pi_y^{\mathrm{tar}} \left( \frac{c_y}{m_y^{\gamma}} + \mathrm{BiasTerm}_y \right).$$

Let SGS allocate $s_y$ seed samples and $g_y$ global-stage samples to class $y$, so $m_y = s_y + g_y$. Then

$$\mathcal{B}_A^{\mathrm{SGS}}(\pi^{\mathrm{tar}}) = \sum_y \pi_y^{\mathrm{tar}} \left( \frac{c_y}{(s_y + g_y)^{\gamma}} + \mathrm{BiasTerm}_y \right)$$

$$= \sum_y \pi_y^{\mathrm{tar}} \left( \frac{c_y}{s_y^{\gamma}} + \mathrm{BiasTerm}_y \right) - \sum_y \pi_y^{\mathrm{tar}} c_y \left( s_y^{-\gamma} - (s_y + g_y)^{-\gamma} \right).$$

The second term is nonnegative because $\gamma > 0$ and $s_y + g_y \geq s_y$, and it is strictly positive for any class with $g_y > 0$. If the seed stage guarantees $s_y \geq b$ for every class, then

$$\sum_y \pi_y^{\mathrm{tar}} \left( \frac{c_y}{s_y^{\gamma}} + \mathrm{BiasTerm}_y \right) \leq \sum_y \pi_y^{\mathrm{tar}} \left( \frac{c_y}{b^{\gamma}} + \mathrm{BiasTerm}_y \right).$$

Therefore,

$$\mathcal{B}_A^{\mathrm{SGS}}(\pi^{\mathrm{tar}}) \leq \sum_y \pi_y^{\mathrm{tar}} \left( \frac{c_y}{b^{\gamma}} + \mathrm{BiasTerm}_y \right) - \sum_y \pi_y^{\mathrm{tar}} c_y \left( s_y^{-\gamma} - (s_y + g_y)^{-\gamma} \right).$$

This shows that the seed stage gives a prior-robust baseline guarantee, and the global stage refines this baseline through an explicit residual gain.

## A.6. Second-Moment Bound for Hard-Label Reweighting

Suppose the pruned subset induces class prior $\rho$ but the target prior is $\pi^{\text{tar}}$. Hard-label training corrects this mismatch by assigning class weight $a_y = \pi_y^{\text{tar}}/\rho_y$ to samples from class $y$. Let $g(x, y)$ be the unweighted per-sample gradient and assume $\|g(x,y)\| \leq G$. A stochastic gradient drawn from the pruned subset and reweighted toward $\pi^{\text{tar}}$ is

$$\hat{g} = a_y g(x, y), \qquad y \sim \rho.$$

Then

$$\mathbb{E}\|\hat{g}\|^2 = \sum_y \rho_y a_y^2 \mathbb{E}\left[\|g(x,y)\|^2 \mid y\right] \leq G^2 \sum_y \rho_y \left(\frac{\pi_y^{\text{tar}}}{\rho_y}\right)^2 = G^2 \sum_y \frac{(\pi_y^{\text{tar}})^2}{\rho_y}.$$

By Cauchy's inequality,

$$\sum_y \frac{(\pi_y^{\text{tar}})^2}{\rho_y} \geq \frac{\left(\sum_y \pi_y^{\text{tar}}\right)^2}{\sum_y \rho_y} = 1,$$

with equality if and only if $\rho = \pi^{\text{tar}}$. Thus, the second-moment bound is minimized when the pruned prior matches the target prior, and grows with prior mismatch.

## A.7. Proof of Proposition 4.1

*Proof.* Fix a subset $S$ and weights $w$ with $w_i > 0$ for all $i \in S$. For brevity write

$$T_i(\cdot) := T(\cdot \mid x_i) \in \Delta^{|\mathcal{Y}|-1}, \qquad f_i(\cdot) := f_\theta(\cdot \mid x_i) \in \Delta^{|\mathcal{Y}|-1}.$$

Recall that the (categorical) cross-entropy is

$$\text{CE}(T_i, f_i) := -\sum_{y \in \mathcal{Y}} T_i(y) \log f_i(y),$$

with the usual convention that if $T_i(y) > 0$ and $f_i(y) = 0$ then $\text{CE}(T_i, f_i) = +\infty$.

Using the standard decomposition of cross-entropy into entropy plus KL divergence, for each $i \in S$ we have

$$\text{CE}(T_i, f_i) = -\sum_y T_i(y) \log f_i(y)$$
$$= -\sum_y T_i(y) \log T_i(y) + \sum_y T_i(y) \log \frac{T_i(y)}{f_i(y)}$$
$$=: H(T_i) + \text{KL}(T_i \| f_i),$$

where $H(T_i)$ is the Shannon entropy and $\text{KL}(\cdot\|\cdot)$ is the Kullback–Leibler divergence. By Gibbs' inequality, $\text{KL}(T_i\|f_i) \geq 0$, with equality if and only if $f_i(\cdot) = T_i(\cdot)$ (as distributions on $\mathcal{Y}$).

Plugging this into the weighted KD objective (9) yields

$$\hat{\mathcal{L}}_{\text{KD}}^{S,w}(\theta) = \sum_{i \in S} w_i \, \text{CE}(T_i, f_i)$$
$$= \sum_{i \in S} w_i \, H(T_i) + \sum_{i \in S} w_i \, \text{KL}(T_i \| f_i).$$

The first term $\sum_{i \in S} w_i H(T_i)$ does not depend on $\theta$. The second term is a weighted sum of nonnegative quantities, hence

$$\hat{\mathcal{L}}_{\text{KD}}^{S,w}(\theta) \geq \sum_{i \in S} w_i \, H(T_i),$$

and equality holds if and only if $\text{KL}(T_i\|f_i) = 0$ for every $i \in S$, i.e., if and only if $f_\theta(\cdot \mid x_i) = T(\cdot \mid x_i)$ for all $i \in S$.

By assumption, the student is expressive enough to interpolate the teacher on $S$, so there exists at least one $\theta$ satisfying $f_\theta(\cdot \mid x_i) = T(\cdot \mid x_i)$ for all $i \in S$. For any such interpolating $\theta$, the KL terms vanish and the lower bound is achieved; therefore every interpolating solution is a global minimizer of $\hat{\mathcal{L}}_{\text{KD}}^{S,w}$. Since the argument holds for any choice of strictly positive weights $w$, the claim follows. $\square$

# B. Additional Experimental Setup

## B.1. Model Architecture

We use PointNeXt-S and PointVector-S. For MeshMAE and PointMAE, we use their default architecture when processing corresponding datasets. The PointMAE and MeshMAE models directly reuse their official checkpoints. Table 6 reports the performance of models trained by us on different datasets.

## B.2. Training Strategy

Our training pipeline is conducted in three phases. Unless otherwise specified, the main experiments use random seed 42; the multi-seed evaluation in Sec. C.1 repeats representative ShapeNet55 settings with seeds 42, 0, and 1. We strictly follow the data augmentation protocols outlined in the original papers for each model.

*Table 6.* Cross-dataset evaluation across 3D backbones. Overall accuracy (OA) and mean accuracy (mAcc) on ShapeNet55, ModelNet40, and ScanObjectNN.

| | ShapeNet55 | | ModelNet40 | | ScanObjectNN | |
|---|---|---|---|---|---|---|
| Model | OA | mAcc | OA | mAcc | OA | mAcc |
| PointNeXt-S | 91.11 | 83.47 | 93.15 | 90.60 | 87.20 | 85.80 |
| PointNet++ | 91.08 | 82.83 | 92.95 | 90.34 | 86.12 | 84.37 |
| PointVector-S | 91.00 | 82.96 | 90.15 | 85.31 | 87.37 | 85.64 |

**Initial Model Training.** The base model is trained for 300 epochs with a batch size of 48. We utilize the AdamW optimizer with an initial learning rate of $1 \times 10^{-4}$ and a weight decay of $0.05$. The learning rate is managed by a cosine annealing scheduler with 2 epochs of linear warmup. Notably, we apply label smoothing of $0.2$ throughout this phase.

**Class-Balanced Classifier Retraining.** In this phase, we freeze the encoder and retrain the reinitialized classification head for 100 epochs. We employ a UniformClassSampler and mixed-precision (bfloat16) training. The learning rate is adjusted to $1.7 \times 10^{-4}$, while the weight decay remains $0.05$ and the warmup period stays at 2 epochs. Consistent with the initial phase, the label smoothing parameter is maintained at $0.2$.

**Knowledge Distillation.** The student model is trained from scratch on the pruned dataset for 300 epochs, reusing the optimizer configuration from the initial phase. For the distillation parameters, we set the temperature $\tau = 5$ and the knowledge distillation weight $\alpha = 0.8$. Regarding Relational Knowledge Distillation (RKD), we use a distance weight $\lambda_d = 50$ and an angle weight $\lambda_a = 100$, applied with an overall scaling factor of $\lambda = 0.1$.

# C. Additional Results

## C.1. Multi-Seed Evaluation

To further validate the stability of our observations, we repeat representative ShapeNet55 experiments with three random seeds (42, 0, 1). Table 7 reports the mean and standard deviation. The results support the same conclusions as the main experiments: embedding-based FL-RBF is a strong global selector, CSL and EGD improve the base selector, and SGS provides a controllable allocation mechanism through $K$. In particular, smaller $K$ favors OA, while larger $K$ consistently shifts the subset toward stronger mAcc, matching the intended role of $K$ as a preference-control parameter.

*Table 7.* Multi-seed evaluation on ShapeNet55 with PointNet++. Results are reported as mean±std over three seeds (42, 0, 1), with each entry formatted as OA/mAcc.

| Setting | $m = 550$ | $m = 1100$ | $m = 2200$ |
|---|---|---|---|
| Native FL-RBF | 83.4±0.4/65.2±0.8 | 85.8±0.3/70.2±0.6 | 87.2±0.2/74.2±0.1 |
| + CSL + EGD | 84.6±0.4/68.4±0.7 | 87.5±0.2/73.3±0.5 | 88.7±0.2/78.3±0.3 |
| + CSL + EGD + SGS ($K = 0.2$) | **85.2±0.1**/72.3±0.3 | **87.7±0.2**/74.8±0.2 | **88.9±0.2**/79.0±0.2 |
| + CSL + EGD + SGS ($K = 0.4$) | 84.5±0.0/**73.8±0.3** | 87.2±0.2/**77.8±0.6** | 88.5±0.2/**80.7±0.4** |

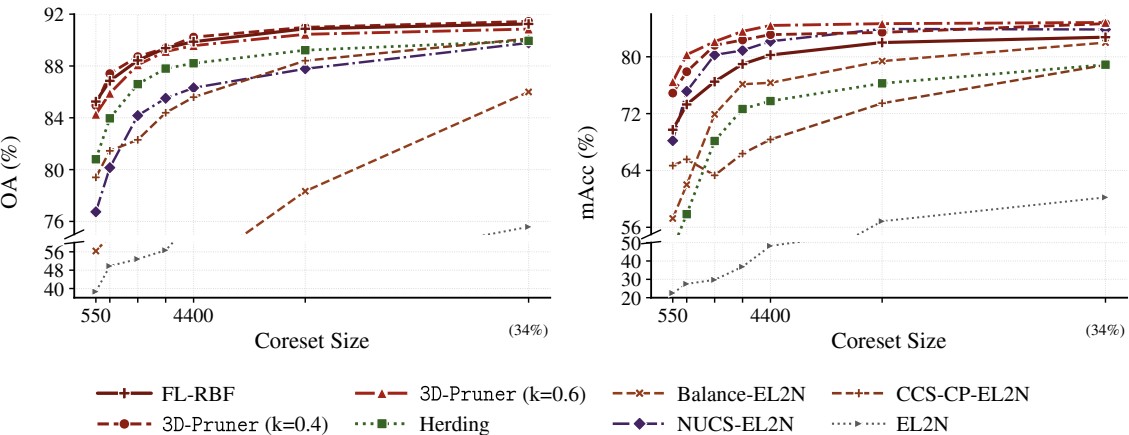

*Figure 9.* Additional results on relaxed budgets on ShapeNet55. The gap on OA becomes smaller given larger budget, yet our `3D-Pruner` remains the best on mAcc.

## C.2. Additional Results on Different Architectures

Tab. 9 presents results on PointNeXt and PointVector, demonstrating that our method generalizes beyond the architectures in the main paper. Tab. 8 shows results on MeshMAE, a more modern mesh-based architecture, further confirming the robustness of our approach. These findings support the broad effectiveness of our proposed shared principles.

*Table 8.* Comparison of different pruning methods on ModelNet40, MeshMAE. The pruning methods are selected from the most competitive ones based on the performance on the MeshNet.

| | | ModelNet40 | | | | | |
| | | $m = 400$ | | $m = 800$ | | $m = 1200$ | |
| **Metric Type** | **Selection Policy** | OA | mAcc | OA | mAcc | OA | mAcc |
|---|---|---|---|---|---|---|---|
| **Scalar Scores** | *Stratified Sampling* | | | | | | |
| | Balance-EL2N | 60.60 | 56.71 | 76.83 | 74.72 | 82.85 | 80.68 |
| | NUCS-EL2N | 57.06 | 52.99 | 72.89 | 70.68 | 81.27 | 78.40 |
| | CCS-CP-EL2N | 68.25 | 57.17 | 78.83 | 70.07 | 83.12 | 75.21 |
| **Vector Emb.** | *Global Selection* | | | | | | |
| | K-Center | 59.00 | 58.16 | 79.14 | 74.63 | 83.17 | 79.45 |
| | FL-RBF | 76.45 | 68.71 | 85.43 | 80.64 | 87.95 | 84.05 |
| | **SGS (Ours)** | | | | | | |
| | FL-RBF (K=0.4) | 75.87 | 70.50 | 85.45 | 80.68 | 88.27 | 85.01 |
| | $\Delta$ vs FL-RBF | -0.58 | +1.79 | +0.02 | +0.04 | +0.32 | +0.96 |
| | FL-RBF (K=0.6) | 76.85 | 71.19 | 85.87 | 80.89 | 88.19 | 85.22 |
| | $\Delta$ vs FL-RBF | +0.40 | +2.48 | +0.44 | +0.25 | +0.24 | +1.17 |

## C.3. Additional Results on Relaxed Budgets

Fig. 9 presents comparisons under more relaxed budgets (retaining up to 40% of the original data). We select the best-performing pruning methods and further increase the budget to compare them. Results validate the effectiveness of `3D-Pruner` at larger pruning ratios; nevertheless, we observe that the preference steering induced by different $K$ values diminishes under higher budgets, particularly at 34%. As analyzed in the main paper, ShapeNet55 contains fewer than 5% few-shot cases. However, larger budgets increase the number of many-shot classes, making the prior mismatch term more dominant. This explains why our method remains competitive ***on mAcc*** under this setting, as it mitigates this effect through calibrated teacher soft-labels and geometric embedding distillation.

*Table 9.* Additional results on PointNeXt and PointVector.

| | PointNeXt-S | | | | | | PointVector-S | | | | | |
| | $m = 550$ | | $m = 1100$ | | $m = 2200$ | | $m = 550$ | | $m = 1100$ | | $m = 2200$ | |
| **Method** | OA | mAcc | OA | mAcc | OA | mAcc | OA | mAcc | OA | mAcc | OA | mAcc |
|---|---|---|---|---|---|---|---|---|---|---|---|---|
| Loss | 71.32 | 31.82 | 75.90 | 41.46 | 78.38 | 40.01 | 71.75 | 28.02 | 74.71 | 34.17 | 76.35 | 35.37 |
| GradNorm | 76.75 | 37.96 | 79.36 | 43.69 | 82.25 | 53.49 | 80.25 | 49.30 | 82.76 | 56.60 | 84.31 | 60.24 |
| Entropy | 72.01 | 30.96 | 74.98 | 34.71 | 78.91 | 41.62 | 71.99 | 31.82 | 75.93 | 38.64 | 78.05 | 42.24 |
| EL2N | 71.71 | 31.69 | 76.57 | 36.99 | 78.04 | 41.45 | 70.47 | 26.15 | 74.42 | 30.29 | 78.29 | 37.66 |
| DRoP | 50.11 | 59.31 | 59.05 | 68.10 | 64.36 | 72.57 | 42.30 | 56.77 | 52.87 | 63.24 | 59.56 | 69.99 |
| Balance-EL2N | 80.51 | 74.92 | 83.55 | 78.59 | 85.90 | 82.18 | 80.23 | 75.34 | 82.11 | 76.56 | 84.92 | 80.22 |
| NUCS-EL2N | 78.01 | 71.94 | 81.76 | 76.68 | 85.48 | 80.94 | 77.63 | 71.99 | 81.53 | 77.58 | 84.60 | 80.56 |
| CCS-CP-EL2N | 79.80 | 52.77 | 82.40 | 59.20 | 83.90 | 59.74 | 79.44 | 53.41 | 81.56 | 57.67 | 83.96 | 62.23 |
| Herding | 83.05 | 59.61 | 85.06 | 64.00 | 87.44 | 72.21 | 81.93 | 57.13 | 84.97 | 67.28 | 87.10 | 71.25 |
| K-Center | 59.77 | 60.96 | 76.38 | 69.32 | 85.99 | 76.04 | 56.24 | 62.95 | 69.47 | 67.99 | 84.21 | 76.92 |
| FL-RBF | 82.51 | 74.35 | 83.45 | 77.78 | 86.10 | 80.92 | 83.13 | 74.20 | 85.41 | 76.45 | 86.20 | 80.32 |
| **3D-Pruner (Ours)** | | | | | | | | | | | | |
| FL-RBF (K=0.4) | **85.50** | 75.75 | **86.80** | 78.80 | **88.72** | 81.02 | **84.78** | 76.11 | **87.41** | 79.48 | **88.91** | 82.96 |
| FL-RBF (K=0.6) | 83.68 | **76.51** | 86.04 | **80.54** | 88.35 | **83.51** | 84.07 | **76.46** | 86.29 | **80.43** | 88.23 | **82.98** |

