# OpenReview forum: "Exploring 3D Dataset Pruning"
_ICML.cc/2026/Conference — ICML 2026 regular_

### Official Review · Reviewer_5rpR · 2026-03-02

**Soundness:** 4
**Presentation:** 3
**Significance:** 4
**Originality:** 4
**Overall Recommendation:** 4
**Confidence:** 4

**Summary:**

The paper addresses 3D dataset pruning under inherent class imbalance by proposing the Steering upon Shared Principle
Pruning, which integrates safety-aware global selection and a distillation pipeline with Calibrated Soft Labels and Embedding Geometry Distillation. Guided by quadrature error decomposition, the framework aims to ensure a stable performance floor and flexible prior preferences. Its key contribution is developing this integrated framework to improve pruning performance under imbalance, validated by experiments showing the proposed method outperforms baselines across 3D datasets, models and modalities.

**Compliance With Llm Reviewing Policy:**

Affirmed.

**Final Justification:**

The paper has key strengths that align with initial positive evaluations. The authors’ rebuttal effectively addresses most initial weaknesses. Key evaluation dimensions remain strong, with result unchanged.

**Key Questions For Authors:**

1. Could you clarify whether there is an explanation of the pruning function of SGS itself, rather than the combined function of SGS and distillation? If possible, please further analyze the results of SGS and FL-RBF both with and without CSL and EGD to further verify the independent value of SGS in pruning.
2. Why is RKD suitable for 3D data? Provide a geometric interpretation of its effectiveness in 3D embedding geometry signal transfer and explain any 3D adaptations.
3. Table 4 indicates a slight decrease in OA when using SGS; could you analyze the potential reasons for this decrease?

**Limitations:**

The authors inadequately discuss limitations: they ignore SGS performance across different 3D data, omit manual K tuning, and fail to confirm whether CSL and EGD distillation are the decisive factor for performance improvement. Manual K tuning should be improved to adaptive selection in the future. The authors should acknowledge these limitations and propose corresponding future improvements.

**Strengths And Weaknesses:**

Strengths:
1. A clear quadrature error decomposition framework is presented in the paper, with comprehensive experiments that cover 3D datasets, models and baselines.
2. Following the strategy of establishing robust base principles before adapting preferences via a steering wrapper, the paper is well-structured; its narrative is logical, and figures effectively illustrate key claims.
3. Focusing on high-compression regimes to distinguish pruning from head-class downsampling, the paper addresses an underexplored gap in imbalanced 3D dataset pruning, and validates effectiveness on point cloud and other modalities to deliver practical value.
4. For 3D data, the paper combines geometric embedding selection and robust distillation, and SGS integrates stratified seeding and global embedding selection to mitigate few-shot class under-coverage; the paper also analyzes embedding signal superiority over scalar signals.
Weaknesses：
1. Quantitative validation of quadrature error decomposition and separate measurement of representation error and prior-mismatch bias reduction need to be more comprehensive and detailed.
2. Ablation experiments to isolate SGS contribution need further refinement and elaboration. SGS has not been tested in sufficient detail without CSL and EGD, making its independent value unverifiable.
3. The paper applies Relational Knowledge Distillation to EGD for embedding geometry signal transfer, but explanations for its suitability for 3D data and geometric interpretation should be more thorough and clear.
4. SGS requires manual tuning of steering parameter K, and its automatic selection mechanisms need to be further improved and supplemented to facilitate practical deployment.
5. SGS combines existing stratified seeding and global geometric selection, and its new mechanisms should be more elaborate and comprehensive; it merely applies them incrementally to 3D data. The safety floor strategy is also common and lacks innovative improvements.

---

> ### Author Rebuttal · Authors · 2026-03-31
>
> We thank the reviewer for the thorough and insightful feedback, which will help us improve the paper. We address each point below:
>
> >**W1/W2/Q1.** Validation of error decomposition needs more detail; SGS not tested without CSL/EGD, making its independent value unverifiable. Isolate SGS's pruning function from distillation.
>
> **A1.** Since W1, W2, and Q1 are naturally addressed through a unified ablation, we respond together. We isolate SGS without CSL or EGD (Tab. R1), directly demonstrating SGS's contribution. We additionally ablate CSL+EGD (Tab. R2) for full ablation.
>
> *Tab. R1. SGS without distillation on ShapeNet55 (PointNet++).*
> |Method|m=550 (OA/mAcc)|m=2200 (OA/mAcc)|
> |-|-|-|
> |FL-RBF|83.4±0.4/65.2±0.8|87.2±0.2/74.2±0.1|
> |SGS(K=0.2)|83.6±0.3/68.3±1.1|87.4±0.3/76.7±0.3|
> |SGS(K=0.4)|83.1±0.5/71.9±0.9|86.9±0.3/78.6±1.3|
> |SGS(K=0.6)|81.7±0.6/73.7±0.3|86.1±0.1/79.6±1.0|
> |SGS(K=1.0)|76.1±0.9/74.9±1.0|82.8±0.1/80.1±0.4|
>
> *Tab. R2. Effect of CSL+EGD and SGS.*
> |Setting|m=550 (OA/mAcc)|m=2200 (OA/mAcc)|
> |-|-|-|
> |FL-RBF|83.4±0.4/65.2±0.8|87.2±0.2/74.2±0.1|
> |+CSL+EGD|84.6±0.4/68.4±0.7|88.7±0.2/78.3±0.3|
> |+CSL+EGD+SGS(K=0.2)|**85.2**±0.1/72.3±0.3|**88.9**±0.2/79.0±0.2|
> |+CSL+EGD+SGS(K=0.4)|84.5±0.0/**73.8**±0.3|88.5±0.2/**80.7**±0.4|
>
> >**W3/Q2.** RKD's suitability for 3D data and geometric interpretation need more thorough explanation; any 3D adaptations?
>
> **A2.** We clarify that our draft did not clearly distinguish raw 3D geometry from the geometry of the learned embedding space. RKD is not uniquely designed for 3D data; rather, it is suitable because 3D encoders learn embeddings organized by geometric primitives shared across head and tail classes. After pruning, the teacher manifold becomes sparse, so matching only logits preserves decision boundaries but not the internal topology of the embedding space. RKD addresses this by aligning pairwise distances (preserving neighborhood relations) and triplet angles (preserving higher-order orientation), retaining the internal structure of class manifolds under pruning-induced sparsity. We do not introduce a 3D-specific RKD formulation; the 3D specificity arises from applying standard RKD to geometry-aware point-cloud embeddings alongside our calibrated teacher. We will make this distinction explicit in the revision.
>
> >**W4.** SGS requires manual tuning of $K$; automatic selection needed for practical deployment.
>
> **A3.** A small $K$ (e.g., 0.1–0.2) already yields joint improvements in both OA and mAcc (Fig. 7), requiring no careful tuning. Beyond this range, increasing $K$ trades OA for mAcc along a smooth, interpretable curve (Tab. R1/R2), allowing the practitioner to select an operating point according to their preference. This makes $K$ a Pareto-front control knob rather than a sensitive hyperparameter. Linking $K$ to dataset-level properties for automatic selection is a natural extension for future work; discussion added to the revision.
>
> >**W5.** SGS combines existing stratified seeding and global geometric selection; new mechanisms need more elaboration. Safety floor strategy is common.
>
> **A4.** We would like to clarify the nature of the contribution. We are, to our best knowledge, the first to study dataset pruning for 3D point clouds and to frame the OA/mAcc trade-off as a concrete problem. The core novelty lies in the prior-aware reformulation that characterizes this tension through representation error and prior mismatch, providing the unifying logic that determines how and why the components are composed: the per-class safety floor bounds representation error while global geometric further refines it. This compositional rationale, validated across three 3D benchmarks, multiple architectures, and extended to other modalities, distinguishes our framework from ad hoc combination of known techniques. Additional formal analysis is provided in our response to Reviewer V9Ex (A1–A3).
>
>
> >**Q3.** Tab. 4 shows slight OA decrease with SGS; analyze potential reasons.
>
> **A5.** As detailed in A3, this is the expected Pareto behavior of $K$: at $K\le 0.2$ both metrics improve jointly, while larger $K$ trades OA for mAcc. The reductions in Tab. 4 specifically arise because (1) CSL and EGD are enabled for *all* methods, isolating SGS as the variable, and (2) $K$=0.4/0.6 fall in the trade-off regime. Tab. R1/R2 confirm this is a specific operating point on the Pareto front, not a deficiency of SGS.
>
> >**Q4.** Limitations discussion insufficient: SGS across different 3D data, manual K tuning, and whether CSL/EGD are the decisive factor.
>
> **A6.** These concerns are substantively addressed in this rebuttal: SGS's independent contribution is
> quantified in Tab. R1/R2 (A1); $K$'s behavior across regimes is clarified in A3 and A5; and Tab. 4 already controls for CSL/EGD by applying them to *all* methods, isolating SGS. Broader 3D coverage and
> adaptive $K$ selection would further strengthen the practical scope of the work, and we will clarify this in the revision.

---

> > ### Author Rebuttal · Reviewer_5rpR · 2026-04-02
> >
> > The authors have provided comprehensive and convincing responses to all my concerns with detailed ablation analyses and clear explanations. All issues are fully resolved, and I will maintain my original score.

---

> > > ### Author Response · Authors · 2026-04-04
> > >
> > > We sincerely thank the reviewer for the thorough evaluation and constructive feedback, which have been very helpful in improving the paper. We are glad that the responses have fully addressed the concerns, and all improvements will be incorporated into the revised manuscript.

---

### Official Review · Reviewer_bMzp · 2026-03-08

**Soundness:** 3
**Presentation:** 2
**Significance:** 3
**Originality:** 3
**Overall Recommendation:** 5
**Confidence:** 3

**Summary:**

The paper proposed a novel data pruning approach for imbalanced datasets, focusing on 3D data. The paper argues that the commonly used metric, overall accuracy, and mean accuracy have conflicting goals for these distributions since they encode different priors, such that for each metric, the optimal set differs.
The proposed S^2P^2 method addresses this with a teacher-student rational knowledge distillation and a geometry-aware selection. The paper evaluated the new methods extensively on different models, modalities, and datasets.

**Compliance With Llm Reviewing Policy:**

Affirmed.

**Final Justification:**

The authors have provided additional experiments, which have been one of my core concerns.
In addition, the authors have provided details on how to improve the other weakness in an updated version. Given this details I increase my score.

**Key Questions For Authors:**

1) Interestingly, S^2P^2 usually outperforms in both metrics. I would have assumed just a Pareto-optimal behaviour, and a class method outperforming in OA. Any idea why?
2) Does this not violate the dilemma, since also in Table 4, the same k shows the best performance in both metrics?

3) Why in Table 3, FL-RBF is listed with different k under SGS? Is FL-RBF part of the proposed method?

4) Algorithm 1: small k capital K mismatch?

5)CE → Crossentropy?

6) EQ 10 diract of y so hard label?
7) Why do you need a Dirac function? Samples are already discrete.
8) Why “w” have size m-1 and S has size m?

**Limitations:**

The authors should include a limitations section that discusses potential failure cases of the method.
The negative impact has been stated.

**Strengths And Weaknesses:**

## Strengths

* The paper investigates pruning for an imbalanced dataset, highlighting the dilemma posed by the goal metric.
* The paper provides an evaluation of numerous datasets, models, and modalities.
* The proposed method is effective.





## Major Weaknesses


**Clearity of Theory and Methodology**

* Many variables are not introduced in Section 3, making the theory hard to understand. While it is clear that the section should showcase the discrepancy between mean and overall accuracy assuming different class distributions, the deviation is too high-level. While proofs are listed in the appendix, they are not referenced in section 3. They should be properly referenced to understand Section 3 on its own, with the provided references, and the required transformation should be in the main paper.
What are g, theta, and delta ? Are theta the model's weights? State it explicitly.
* From the description, it sounds like imbalance is predominantly present in 3D modalities, which is often true across many real-world tasks and modalities.  I cannot see why 3D should differ from an unbalanced 2D dataset, and would be interested to see an evaluation of unbalanced CIFAR-100 or a similar dataset.
* The entanglement between the appendix and the main paper should be generally improved, providing efficient and referenced lookups for more details or results.
* The final equation for SGS is missing, and how the importance is estimated.


**Experiments and Baseline**

* The paper should report the pruning percentage instead of the absolute dataset size.
* Many recent and standard pruning baselines [A-D] are missing in the paper. Any reason for that?
* Single-seeded evaluation: The experiments are conducted with only one seed (42), which does not provide statistical significance.
* Random Selection baseline is missing.
* The differences between k’s seem to collapse at lower pruning rates (Fig. 7). I would be interested to see this evaluated.
* Discussion of results and the relation to the dilemma stated in section 3 is missing (see question).
* It is unclear how the baselines are exactly used/combined. This should be stated in the appendix e.g. NUCS-EL2N.

## Minor Weaknesses

* Table 3  Loss should be bold for R2.
* B is used as Bound and B term.
* Table 2 is unreferenced.
* The Usage of S^2P^2 and SGS seems to be inconsistent. To my understanding, SGS is the selection score, and S^2P^2 is the full approach combining SGS with teacher-student training.

I am willing to reconsider my score if my main concerns can be addressed.

[A] Coverage-centric Coreset Selection for High Pruning Rates; Haizhong Zheng, Rui Liu, Fan Lai, Atul Prakash; ICLR 2023

[B] Moderate Coreset: A Universal Method of Data Selection for Real-world Data-efficient Deep Learning;
Xiaobo Xia, Jiale Liu, Jun Yu, Xu Shen, Bo Han, Tongliang Liu; ICLR 2023

[C] Spanning Training Progress: Temporal Dual-Depth Scoring (TDDS) for Enhanced Dataset Pruning; Xin Zhang, Jiawei Du, Yunsong Li, Weiying Xie, Joey Tianyi Zhou; CVPR 2024

[D] Identifying mislabeled data using the area under the margin ranking; Geoff Pleiss, Tianyi Zhang, Ethan R. Elenberg, Kilian Q. Weinberger; Neurips 2020

---

> ### Author Rebuttal · Authors · 2026-03-31
>
> We thank the reviewer for the constructive comments. We address each point below:
>
> >**W1/W3.** Variables ($\theta,\delta,g,\mathcal{G}$) lack definitions; appendix proofs not forward-referenced.
>
> **A1.** All variables are now defined at first use in Sec. 3 with forward references to Appendix proofs. Specifically: $\theta$ = trainable parameters; $\delta$ = Dirac delta for the pruned empirical distribution; $\mathcal{G}=\{\ell_\theta(\cdot):\theta\in\Theta\}$ is the function class, and $g$ is one particular instance in $\mathcal{G}$.
>
> >**W2.** Class imbalance not unique to 3D; include imbalanced 2D evaluation.
>
> **A2.** Our point is not that long-tail is unique to 3D, but that many canonical 2D long-tailed benchmarks adopt balanced or near-balanced evaluation, under which the OA/mAcc gap is smaller. In our 3D benchmarks, the class prior *persists to evaluation*. We construct imbalanced 2D evaluation splits following the training prior; the same tension emerges and our method stays effective. (Std omitted for space; multi-seed results with full std in A4/A6.)
> *CIFAR-100-LT (imbalance ratio 10, ResNet-34), OA/mAcc.*
> ||m=1000 (94.9%)|m=3000 (84.7%)|
> |--|--|--|
> |AUM [D]|19.4/12.6|33.5/22.0|
> |Moderate [B]|24.5/17.6|45.3/33.8|
> |TDDS [C]|15.3/14.5|28.0/27.1|
> |CCS-CP EL2N [A]|19.5/20.3|39.7/40.3|
> |Herding|23.5/17.8|44.2/35.6|
> |K-Center|16.3/17.2|32.0/33.2|
> |FL-RBF|17.1/17.2|33.3/33.3|
> |**S$^2$P$^2$ (K=0.4)**|**27.8**/**24.5**|**48.2**/**45.4**|
> |**S$^2$P$^2$ (K=0.6)**|**28.0**/**25.1**|**49.5**/**47.1**|
>
> >**W4/Q3.** Final SGS equation missing; how is importance estimated? Why does FL-RBF appear under SGS with different $k$?
>
> **A3.** SGS is a plug-and-play wrapper around any base selector $\phi$; it reuses $\phi$'s own scores. The selection (in Alg. 1) is:
> $$b=\left\lfloor\frac{KB}{|\mathcal{C}|}\right\rfloor,\quad\mathcal{S}=\left(\bigcup_{c\in\mathcal{C}}\phi(\mathcal{D}_c,b,\emptyset)\right)\cup\phi(\mathcal{D},B-b|\mathcal{C}|,\emptyset),$$ with seeded fill-up if overlap occurs.
> FL-RBF appears under SGS as the wrapped base selector; $K$ controls the per-class vs. global split. Unlike NUCS and CCS-CP, SGS accommodates iterative selectors natively.
>
> >**W6/W7/W8.** Several recent baselines [A–D] missing; single seed (42); no random baseline.
>
> **A4.** (W6) CCS [A] is already included as CCS-CP [E]. We have added [B], [C], [D], and Random; results below.
> (W7) All results re-evaluated over 3 seeds.
> (W8) Included below.
> *Additional baselines on ShapeNet55 (PointNet++), OA/mAcc (%).*
> |Method|m=550 (98.7%)|m=2200 (94.8%)|
> |-|-|-|
> |CCS-CP [E]|76.0±0.5/61.8±1.6|79.0±0.3/52.9±3.2|
> |Moderate [B]|74.8±1.1/49.2±5.0|79.4±0.6/59.9±2.2|
> |TDDS [C]|79.5±1.3/51.1±3.9|85.0±0.1/63.7±1.6|
> |AUM [D]|51.8±3.2/22.3±0.9|64.2±0.7/44.4±1.4|
> |Global Random|80.0±0.3/49.4±1.1|85.1±0.4/65.3±2.0|
> |Strat. Random|74.0±1.8/70.9±0.7|81.4±0.6/79.8±0.2|
> |**S$^2$P$^2$ (K=0.4)**|**84.5**±0.0/73.8±0.3|**88.5**±0.2/80.7±0.4|
> |**S$^2$P$^2$ (K=0.6)**|83.6±0.3/**75.3**±0.6|87.7±0.1/**81.8**±0.8|
>
> >**W9.** Differences between $k$'s collapse at lower pruning rates (Fig. 7).
>
> **A5.** The diminishing effect of $K$ on OA is expected: high-frequency classes are covered regardless of allocation. The mAcc effect remains pronounced (M=4400 below); the mAcc spread across $K$ stays wider than OA:
> |K|OA (%)|mAcc (%)|
> |--|--|--|
> |0.0|90.3±0.0|81.3±0.4|
> |0.4|89.8±0.1|83.5±0.3|
> |0.6|89.5±0.2|84.1±0.3|
> |1.0|88.7±0.2|84.0±0.4|
>
> >**W10/Q2.** If the same $k$ wins on both metrics in Tab. 4, does this not contradict the stated trade-off?
>
> **A6.** In Tab. 4, the two metrics are in fact optimized at different $K$ values:
> |$K$|m=550 (OA/mAcc)|m=2200 (OA/mAcc)|
> |-|-|-|
> |0.2|**85.2**±0.1/72.3±0.3|**88.9**±0.2/79.0±0.2|
> |0.4|84.5±0.0/73.8±0.3|88.5±0.2/80.7±0.4|
> |0.6|83.6±0.3/**75.3**±0.6|87.7±0.1/**81.8**±0.8|
>
> >**W11.** Unclear how baselines are used/combined (e.g., NUCS-EL2N).
>
> **A7.** A hyphenated name (e.g., NUCS-EL2N) denotes a class-aware framework + scalar scoring method. Clarified in appendix.
>
> >**Q1.** Why does S²P² outperform on both OA and mAcc simultaneously rather than showing a Pareto trade-off?
>
> **A8.** The joint improvement stems from two sides. On the training side, CSL and EGD optimize a structural likelihood (Eq. (8)) shared across metrics, benefiting both OA and mAcc. On the selection side, geometrically central "core" samples contribute to both head- and tail-class accuracy; SGS at small $K$ retains these shared samples (Eq. (12)) while gently rebalancing toward underrepresented classes.
>
> >**W5/W12/Q4-Q8.** Report pruning ratio %; notation issues (Table 3 bold, B overloaded, S²P²/SGS inconsistency, k/K, CE, Dirac, simplex dimension). Table 2 unreferenced.
>
> **A9.** (W5) Pruning percentages added in all tables.
> (W12/Q4/Q5) All corrected.
> (Q6-Q8) $\Delta^{m-1}$ denotes probability simplex, corrected to $\Delta^m$; Dirac notation simplified to standard cross-entropy.
>
> [A–D] As cited by the reviewer.
>
> [E] Tsai et al., Class-proportional Coreset Selection, ICCV 2025.

---

> > ### Author Rebuttal · Reviewer_bMzp · 2026-04-03
> >
> > The authors provide extensive experiments to address my concerns.
> > Under the assumption that these experiments and improvements will be transferred to an updated version. I increase my score.

---

> > > ### Author Response · Authors · 2026-04-04
> > >
> > > We are very grateful to the reviewer for the thorough and detailed evaluation, which substantially shaped the improvements in our experiments and presentation. We are glad that the additional results have addressed the concerns. All new experiments and corrections will be fully incorporated into the revised manuscript. Thank you again for the time and effort invested in reviewing our work.

---

### Official Review · Reviewer_hyfq · 2026-03-11

**Soundness:** 4
**Presentation:** 3
**Significance:** 4
**Originality:** 3
**Overall Recommendation:** 4
**Confidence:** 3

**Summary:**

This paper studies dataset pruning for 3D recognition under long-tailed class imbalance, where the authors argue that overall accuracy (OA) and mean class accuracy (mAcc) correspond to two different but both meaningful evaluation priors. The paper formulates pruning as a quadrature approximation of population risk and decomposes the induced error into a representation error term and a prior-mismatch bias term, using this view to explain why no single pruned subset is generally optimal for both OA and mAcc. The authors therefore propose $S^2P^2$ which is a framework that builds prior-robust shared principles and then adds a lightweight steering mechanism. The empirical evaluation covers several 3D datasets, architectures, and pruning baselines across point-cloud and mesh settings, and the reported results suggest that the method can improve the performance floor while providing flexibility across different evaluation preferences.

**Compliance With Llm Reviewing Policy:**

Affirmed.

**Final Justification:**

I thank the authors for their detailed and thoughtful response. My concerns have been addressed, and I will retain my original score.

**Key Questions For Authors:**

Q1 regarding W1. It would be beneficial if the authors could clarify more explicitly which component of the contribution they view as the main methodological novelty relative to prior combinations of pruning, imbalance-aware allocation, and distillation. In particular, is the key novelty the prior-aware formulation, the SGS steering wrapper, or the full integration of these components?

Q2 regarding W2. It would be beneficial if the authors could make the connection between the theory and the final SGS algorithm more explicit. For example, can the authors clarify which assumptions behind the per-class approximation result are intended to approximate the behavior of the actual deterministic geometry-based selector, and in what sense the theory should be interpreted as guiding SGS rather than characterizing it tightly?

Q3 regarding W3. It would be beneficial if the authors could report multi-seed results, or mean/std over several runs for a representative subset of the main experiments, especially in the high-compression regime.

Q4 regarding W4. It would be beneficial if the authors could clarify the regimes in which the method provides improvements in both OA and mAcc versus the regimes in which it mainly improves the OA–mAcc trade-off.

**Limitations:**

yes

**Strengths And Weaknesses:**

S1. One of the major strengths is the paper’s conceptual and theoretical organization. By casting pruning as a quadrature approximation of population risk and decomposing the error into representation error and prior-mismatch bias, the paper provides a coherent explanation for why a single subset is generally not optimal for both OA and mAcc. This theory also connects naturally to the method design: KD-based post-pruning training is used to reduce prior mismatch, while geometry-based selection and a minimum safety floor are used to reduce representation error.

S2. The empirical study is fairly comprehensive and convincing. The paper evaluates across multiple 3D datasets, both point-cloud and mesh modalities, several architectures, and a range of pruning budgets. I also appreciated Table 4, where the authors apply their CSL/EGD post-training recipe across baselines; this makes the argument about the value of geometry-based selection signals more credible than a comparison that only swaps in the full proposed system. The signal audit and the steering plots also help support the paper’s main claims in an interpretable way.

S3. The main novelty is not a single algorithmic trick, but rather the prior-aware problem formulation and the way the paper integrates theory, signal auditing, post-pruning distillation, and a simple steering wrapper into one coherent framework. That is a meaningful contribution, especially for a practical problem that is relatively underexplored in the 3D setting.

W1. At the method level, the novelty is meaningful but moderate in terms of raw algorithmic novelty. Several components are standard on their own, including KD, RKD, embedding-space coreset selection, and stratified/global allocation ideas. The contribution is therefore strongest as a principled synthesis and prior-aware reformulation, rather than as a fundamentally new pruning primitive.

W2. The theory is useful and motivating, but some parts do not fully pin down the final algorithm. For example, the key per-class approximation result is presented informally and depends on quantities such as class complexity and a bias term that are not operationalized. In the appendix, the formal bound assumes i.i.d. class-conditional sampling and an empirical measure within each class, whereas the actual method uses a deterministic geometry-based selector. As a result, the theory supports the paper’s intuition, but it does not yet provide a tight characterization of SGS itself.

W3. Experimental robustness could also be stronger. In the appendix, the training pipeline is described as using a fixed random seed of 42, and the main results are presented without variability estimates across multiple seeds. Since pruning under high compression may be sensitive to training and subset-selection randomness, reporting mean/std over several runs would make the empirical conclusions more solid.

W4. Some empirical claims are slightly stronger than what the tables seem to establish. In particular, the paper states that $S^2P^2$ consistently achieves the best performance on both OA and mAcc, but Table 4 shows that the gains are especially clear and consistent on mAcc, while OA is sometimes only comparable and can be slightly lower than FL-RBF in some regimes.

---

> ### Author Rebuttal · Authors · 2026-03-31
>
> We thank the reviewer for the constructive comments. We respond to each concern below.
>
> >**W1/Q1.** Novelty meaningful but moderate; components standard individually; contribution strongest as principled synthesis and prior-aware reformulation. Clarify the main methodological novelty.
>
> **A1.** We thank the reviewer for recognizing the prior-aware reformulation as the central contribution. The contribution begins upstream of the method: we are, to our knowledge, the first to identify the OA/mAcc tension in 3D data pruning and to formalize it as the interaction between representation error and prior mismatch. This decomposition dictates the method design—the seeded floor bounds representation error, the global stage minimizes representation error, and CSL/EGD provide prior-robust distillation. The individual techniques are building blocks, but their selection and integration follow from the theoretical analysis rather than heuristic combination. Details in A2.
>
> >**W2/Q2.** Theory useful and motivating but does not fully pin down algorithm; bound quantities are vague; formal bound assumes i.i.d. while method uses deterministic selector. Make theory–algorithm connection more explicit.
>
> **A2.** We address two aspects: (1) the quantities in the bound are not abstract placeholders, and (2) SGS directly instantiates the bound's structure.
>
> **The bound quantities.** Under a norm-controlled function class [1], if class-$y$ inputs satisfy $\|x\|_2\le R_y$, the network has depth $L$ with $\|W\_\ell\|\_F\le M\_\ell$, and the loss is $L\_{\mathrm{loss}}$-Lipschitz, the class-conditional Rademacher complexity satisfies
> $$\\widehat{\\mathfrak{R}}\_{S\_y}(\\mathcal{G}\_y)\le\frac{c\_y}{2\sqrt{m\_y}},\qquad c\_y=2\,L\_{\mathrm{loss}}(\sqrt{2L\log 2}+1)\,R\_y\prod\_{\ell=1}^{L}M\_\ell.$$
> So $c_y$ is fully determined by data scale and network architecture. The decay rate $\gamma$ in $m_y^{-\gamma}$ is selector-agnostic: i.i.d. sampling gives $\gamma=\tfrac{1}{2}$; structured rules like kernel herding achieve $\gamma=1$ [2]. The appendix instantiates the i.i.d. case, while Eq. (14) is stated at the generality needed to cover broader selectors (see A4 in response to Reviewer V9Ex for supporting experiments).
>
> **From bound to algorithm.** Let $s_y$ and $g_y$ denote the per-class seed and global-stage budgets, so $m_y=s_y+g_y$. If $s_y\ge b$ for every class, the Term-(A) contribution of SGS satisfies:
> $$\\mathcal{B}\_A^{\\mathrm{SGS}}(\pi^{\\mathrm{tar}})\le\\underbrace{\sum\_y\pi\_y^{\\mathrm{tar}}(\frac{c\_y}{b^\gamma}+\\mathrm{BiasTerm}\_y)}\_{\text{seed baseline}}-\\underbrace{\sum\_y\pi\_y^{\\mathrm{tar}}c\_y(s\_y^{-\gamma}-(s\_y+g\_y)^{-\gamma})}\_{\text{global residual gain}}.$$
> Seeding implements the minimum-floor guarantee controlling worst-case per-class error, while the global stage performs residual refinement. Full derivation in revision.
>
> >**W3/Q3.** Fixed seed 42, no variability estimates. Report mean/std over several runs, especially in high-compression regime.
>
> **A3.** We have re-evaluated representative parts of the main experiment, averaged over 3 seeds (42,0,1):
> *ShapeNet55 (PointNet++), mean±std, OA/mAcc (%).*
> |Method|m=550|m=2200|
> |-|-|-|
> ||OA/mAcc|OA/mAcc|
> |Entropy|61.4±0.4/21.1±2.6|73.3±0.9/29.5±1.0|
> |DRoP|46.8±1.5/51.9±3.5|70.2±2.8/69.7±1.9|
> |NUCS|76.2±0.3/69.0±0.4|82.8±0.3/77.7±0.6|
> |CCS-CP|76.0±0.5/61.8±1.6|79.0±0.3/52.9±3.2|
> |Herding|78.9±0.6/51.7±1.2|85.1±0.1/67.2±0.5|
> |K-Center|57.9±2.1/54.6±3.5|83.7±0.9/74.1±0.3|
> |FL-RBF|83.4±0.4/65.2±0.8|87.2±0.2/74.2±0.1|
> |**S$^2$P$^2$ (K=0.4)**|84.5±0.0/73.8±0.3|88.5±0.2/80.7±0.4|
> |**S$^2$P$^2$ (K=0.6)**|83.6±0.3/75.3±0.6|87.7±0.1/81.8±0.8|
>
> >**W4/Q4.** Claims slightly stronger than tables establish; OA gains less consistent than mAcc. Clarify regimes where both improve vs. where method mainly improves the trade-off.
>
> **A4.** We appreciate the careful reading. Both OA and mAcc improve jointly under small $K$ ($K\le 0.2$, Fig. 7) and when CSL+EGD are applied (Fig. 5). As $K$ increases, the method progressively steers the subset toward tail classes, improving mAcc at the cost of OA—this is the intended behavior of $K$ as a Pareto-front control knob. The slight OA reductions in Tab. 4 arise from two factors: (1) CSL and EGD are enabled for *all* methods, so the only variable is SGS; and (2) the $K$ values used (0.4/0.6) are in the regime where SGS slightly trades OA for mAcc. Multi-seed results (3 seeds):
>
> |Setting|m=550 (OA/mAcc)|m=2200 (OA/mAcc)|
> |-|-|-|
> |Native FL-RBF|83.4±0.4/65.2±0.8|87.2±0.2/74.2±0.1|
> |+CSL+EGD|84.6±0.4/68.4±0.7|88.7±0.2/78.3±0.3|
> |+CSL+EGD+SGS(K=0.2)|**85.2**±0.1/72.3±0.3|**88.9**±0.2/79.0±0.2|
> |+CSL+EGD+SGS(K=0.4)|84.5±0.0/**73.8**±0.3|88.5±0.2/**80.7**±0.4|
>
> [1] Golowich, Rakhlin, and Shamir, Size-Independent Sample Complexity of Neural Networks.
>
> [2] Chen, Welling, and Smola, Super-Samples from Kernel Herding.

---

> > ### Author Rebuttal · Reviewer_hyfq · 2026-04-02
> >
> > Thanks for the responses. My concerns have been fully addressed, and I will maintain my score.

---

> > > ### Author Response · Authors · 2026-04-04
> > >
> > > We sincerely thank the reviewer for the careful evaluation and for confirming that the concerns have been fully addressed. The constructive feedback has been valuable in strengthening both the theoretical presentation and the experimental analysis, and we will incorporate all suggested improvements into the revised manuscript.

---

### Official Review · Reviewer_V9Ex · 2026-03-18

**Soundness:** 2
**Presentation:** 2
**Significance:** 2
**Originality:** 1
**Overall Recommendation:** 4
**Confidence:** 4

**Summary:**

The paper studies dataset pruning for 3D recognition datasets under severe class imbalance, focusing on the tension between two widely used evaluation metrics: overall accuracy (OA) and mean class accuracy (mAcc). The authors argue that these metrics correspond to different implicit class priors and therefore induce different optimal pruning strategies. This main claim is to show -- how pruning algorithm can be designed to remain effective across such divergent evaluation priors rather than committing to one metric during subset construction.

**Underlying Main Problem**: study interaction between pruning strategies and evaluation priors in long-tailed 3D datasets.

The paper first proposes a theoretical perspective that casts dataset pruning as a quadrature approximation of population risk. Within this formulation, the authors decompose the error introduced by pruning into two components: representation error, which reflects how well the selected subset approximates the class-conditional distributions, and prior-mismatch bias, which arises when the class distribution of the pruned subset differs from the target evaluation prior. This decomposition motivates a design philosophy where pruning first optimizes aspects that are beneficial across priors and subsequently introduces mechanisms to adjust the subset toward specific evaluation preferences.

Building on this perspective, the authors propose a framework called $$S^2P^2$$ (Steering upon Shared Principle Pruning). The framework contains three main components. First, it addresses prior mismatch during training via knowledge distillation (KD), using teacher soft labels and relational knowledge distillation to preserve structural information about class relationships when training on a pruned subset. Second, it focuses on reducing representation error by selecting samples based on embedding geometry, arguing that geometric diversity in the learned representation space provides a more stable signal than scalar importance metrics such as loss or entropy, which can correlate with class frequency in imbalanced datasets. Third, the method introduces a minimum per-class floor to ensure that rare classes receive sufficient coverage during pruning.

To allow users to balance different evaluation preferences, the authors introduce a steering mechanism called Seeded Global Selection (SGS). SGS combines two sampling modes: stratified seeding, which guarantees minimum coverage for each class, and global geometry-based selection, which prioritizes representative samples from dense regions of the embedding space. A steering parameter controls the proportion of budget allocated to these two stages, effectively allowing the subset to interpolate between balanced and distribution-following selection regimes.

The proposed framework is evaluated on several 3D datasets, including ModelNet40, ScanObjectNN, and ShapeNet55, across multiple architectures such as PointNet++, PointNeXt, and PointMAE. Experiments compare the method against a variety of pruning strategies including loss-based selection, EL2N, entropy, k-center, herding, and imbalance-aware methods. The results suggest that the proposed approach improves the performance floor across both OA and mAcc and enables controllable trade-offs between the two metrics through the steering parameter.

**Compliance With Llm Reviewing Policy:**

Affirmed.

**Final Justification:**

the rebuttal addressed your main concerns, changed your evaluation

**Key Questions For Authors:**

See Weakness.

**Limitations:**

See weakness.

**Strengths And Weaknesses:**

### Strengths

- The paper studies a practically relevant and under-explored setting: 3D dataset pruning under strong class imbalance, where both OA and mAcc are meaningful evaluation targets.
- The framing of OA and mAcc as corresponding to different effective evaluation priors is conceptually useful and gives a coherent explanation for why a single pruned subset may not be optimal for both metrics.
- The proposed pipeline is intuitively well designed, combining geometry-based selection, a minimum per-class floor, distillation-based post-pruning training, and a simple steering mechanism.
- The empirical study is reasonably broad within the target scope, covering multiple 3D datasets, multiple architectures, and both point-cloud and mesh settings.
- The results appear consistently favorable, especially for mAcc, suggesting that the method improves the performance floor in high-compression regimes.
- Although the theory does not fully justify the algorithm, the paper makes a meaningful effort to connect conceptual analysis with practical method design.
- The SGS steering mechanism is simple, interpretable, and practically useful for navigating the OA vs mAcc tradeoff.

## Weakness
**W.1. The theoretical analysis does not directly justify the proposed algorithm**

While the paper provides a useful conceptual framing of dataset pruning, the theory does not directly analyze or justify the proposed algorithm. The analysis studies an idealized pruning problem by decomposing the discrepancy between the full-data distribution and the pruned subset distribution into two terms: representation error and prior mismatch bias. This decomposition gives intuition for why pruning under class imbalance is difficult, but it is not actually connected to the proposed method in a formal way.

More specifically, the theory derives an ideal marginal sample allocation rule that depends on a class complexity term and an assumed approximation-rate exponent. However, the proposed algorithm (S²P²) does not estimate either of these quantities, nor does it implement the allocation rule suggested by the theory. Instead, the method relies on heuristic components such as:
* enforcing a minimum per-class floor
* selecting the remaining samples using embedding-based global selection
* steering the subset distribution via the SGS procedure

The paper does not provide any theorem showing that the SGS procedure approximates the theoretically optimal allocation, minimizes the discrepancy objective studied in the analysis, or achieves any bound on the resulting representation error. As a result, the theory mainly offers post-hoc intuition about the OA–mAcc trade-off, rather than a formal justification of the proposed pruning algorithm.

**W2. The main decomposition is intuitive but technically limited**

The decomposition into representation error and prior-mismatch bias is reasonable and helpful for exposition, but it appears mathematically straightforward rather than technically deep. At a high level, the bound comes from a class-wise decomposition of the discrepancy between the target and pruned distributions, followed by a total-variation bound on the prior mismatch term. Because of this, the theoretical contribution feels more like a clean reformulation of a known type of argument than a genuinely new analytical result. This would be less of an issue if the paper clearly positioned the theory as explanatory. However, the current presentation sometimes overstates the strength of the theoretical contribution relative to what is actually proved.

**W3. Key assumptions are strong and insufficiently justified**

A central step in the analysis is the assumption that the per-class approximation error follows a rate controlled by a class complexity term and a power-law exponent. This assumption is critical because it leads to the class-allocation rule and underlies the discussion of why OA- and mAcc-optimal subsets should differ.

However, the paper does not justify:
* why this rate should hold in the deep learning setting considered
* how the class complexity term should be interpreted or estimated
* whether the exponent is shared across classes or architectures
* whether these assumptions are empirically supported on the 3D benchmarks used

Without additional justification, the key theoretical conclusions depend heavily on an assumption that is introduced in an abstract way and never validated.

**W4 The knowledge distillation proposition is weak in practical relevance**

The claim that knowledge distillation is “weight-robust” is supported by a proposition that assumes the student can perfectly interpolate the teacher on the selected subset. Under that assumption, the weighting of samples does not affect which interpolating solutions are optimal.

This is mathematically true but practically limited. In aggressive pruning settings, the important question is precisely what happens when the student cannot perfectly match the teacher, when optimization is imperfect, or when model capacity is limited. The proposition does not address these practically relevant regimes, so it provides only a weak justification for the paper’s broader claims about robustness to re-weighting.

**W5 The theory abstracts away the actual optimization and representation-learning dynamics**

The theoretical analysis treats pruning through a distribution-approximation lens, which is clean, but it does not capture the actual training dynamics of deep 3D models. In practice, performance depends on:
* the optimization trajectory
* initialization
* model architecture
* representation geometry induced by the full-data teacher
* interactions between subset selection and distillation

None of these are modeled. As a result, the theory may be useful as a high-level lens, but it does not explain why the specific practical design choices in S²P² should work better than alternative heuristics.

**W5. The gains are real but often modest, especially on OA**

The paper presents consistent improvements in several settings, particularly on mAcc, but many of the gains over the strongest geometry-based baselines appear modest, especially for OA. In a number of comparisons, the proposed method seems to improve mAcc more clearly than OA, which is consistent with the paper’s motivation, but it also means the practical advantage over simpler baselines is sometimes incremental rather than decisive.

**W6. The strongest baseline is already very close to the proposed method**

A substantial part of the improvement appears to come from using FL-RBF or another embedding-based selection mechanism together with better post-pruning training. This raises the question of how much of the improvement should be attributed to the paper’s new principles versus the strength of the underlying embedding-based baseline. In other words, the paper may be improving a good existing pipeline with several additional design choices, but the empirical section does not fully disentangle how much novelty is in the selection method itself.

**W7. Limited task diversity**

The evaluation is focused on 3D classification benchmarks, which is aligned with the paper’s scope, but it also limits the strength of the claims. Since the motivating issue is class-imbalanced pruning under multiple evaluation priors, it would be useful to know whether the same principles hold for:
* other 3D tasks such as segmentation or detection
* larger-scale 3D datasets
* non-3D long-tailed settings, if the method is intended to have broader relevance

As it stands, the empirical evidence supports the classification setting considered, but broader claims should be made cautiously.

**W8. Some comparisons may not fully test whether the proposed “principles” are necessary**

The paper argues for a fairly specific combination of principles. However, there are simpler alternatives one might compare against, such as:
* plain class-balanced subsampling plus distillation
* simple two-stage stratified + global sampling without the specific theory framing
* embedding-based selection with a manually tuned per-class quota
* stronger long-tail training baselines on the same selected subsets

Comparisons of this type would help clarify whether the specific proposed framework is necessary, or whether much of the gain can be reproduced with simpler design choices.

**W9. Several claims feel stronger than the evidence supports**

Statements such as being the “first principled exploration” of 3D dataset pruning, or implying a particularly strong form of universal robustness across priors, feel somewhat overstated. The paper does address an interesting and underexplored setting, but many of the individual ingredients are familiar, and the theory is not strong enough to justify broad principled claims in the usual sense.

**W10. There are multiple typos, grammatical issues, and notation inconsistencies**

The manuscript has a noticeable number of writing issues that reduce clarity. Examples include grammatical problems, awkward phrasing, and several likely typos. A few examples from the text include:

* “This view validate that no single subset fits all target priros”
* “the method unfolds in three parts: first, we target … then applies steering wrapper”
* “can resolve” where the sentence appears incomplete
* “confidant and overlap” where the intended phrase appears to be something like “conflict and overlap”
* L369 “The selection is essential broken”
* L 418 “Therm 3.3” instead of “Theorem 3.3”

There are also notation inconsistencies and places where symbols are introduced somewhat abruptly. This is not fatal, but it does make the paper harder to parse than necessary.

---

> ### Author Rebuttal · Authors · 2026-03-31
>
> Thanks for the detailed, constructive comments, which will help us improve the paper. We address each point below:
>
> >**W1/W2.** Theory indirectly justify algorithm; decomposition intuitive but technically limited.
>
> **A1.** We show that two stages of SGS: 1) the seeded floor bounds the representation error (Term A) in the decomposition, independent of class prior, and 2) the global stage provides a strictly positive residual gain.
> Let $\mathcal{B}_A(S;\pi^{\text{tar}}):=\sum_y\pi_y^{\text{tar}}\left(\frac{c_y}{m_y^\gamma}+\text{BiasTerm}_y\right)$ be the residual bound in Eq. (10) after removing Term (B). If $s_y\ge b$ for every class and $m_y=s_y+g_y$, the Term-(A) contribution of SGS satisfies:
> $$\\mathcal{B}\_A^{\text{SGS}}(\pi^{\text{tar}})\le\\underbrace{\sum_y\pi_y^{\text{tar}}\left(\frac{c_y}{b^\gamma}+\text{BiasTerm}_y\right)}\_{\text{seed baseline}}-\\underbrace{\sum_y\pi_y^{\text{tar}}c_y\left(s_y^{-\gamma}-(s_y+g_y)^{-\gamma}\right)}\_{\text{global residual gain}}.$$
> The first term gives a uniform per-class floor guarantee, without specific quota design; the second is strictly positive whenever $g_y>0$, confirming the global stage refines the seed baseline. Full proof in revision.
>
> >**W3.** Key assumptions strong, insufficiently justified.
>
> **A2.** The two quantities in Eq. (14) are not abstract placeholders: $c_y$ is determined by data scale and network architecture under a norm-controlled function class [1], and $\gamma$ in $m_y^{-\gamma}$ encompasses known rates ($\gamma=1/2$ for i.i.d., $\gamma=1$ for herding [2]). The explicit bound and its connection to the SGS algorithm are detailed in A2 of our response to Reviewer hyfq.
>
> >**W4/W5.** KD weak in practical relevance; theory abstracts away learning dynamics.
>
> **A3.** W4 and W5 raise the same question: do the guarantees extend beyond the optimization target to actual training dynamics?
> We show that hard-label reweighting requires class weights $a_y=\pi_y^{\text{tar}}/\rho_y$ to correct prior mismatch, inflating second moment:
> $$\mathbb{E}\|\hat{g}\|^2\le G^2\sum_y\frac{(\pi_y^{\text{tar}})^2}{\rho_y}\\;\ge\\;G^2,$$
> with equality iff $\rho=\pi^{\text{tar}}$. This is precisely the dynamics-level fragility W5 raises.
> Under KD (our CSL and EGD), prior correction is absorbed into the teacher’s soft targets rather than explicit class weighting. Prop. 4.1 further shows that KD minimizers are invariant to reweighting. Hence, compared with hard-label reweighting, KD avoids this weight-induced variance-inflation mechanism and is more robust to prior shifts in practice.
>
> >**W6/W7.** Gains real but often modest on OA; embedding baseline already strong.
>
> **A4.** We appreciate this question. The choice of embedding-based signals is not incidental: Sec. 4.1 compares embedding- and classification-based sources motivating this design. As S$^2$P$^2$ is a plug-and-play module, we apply it ($K=0.4$) on top of other embedding-based selectors (aside from FL-RBF):
> |Method|m=550 (OA/mAcc)|m=2200 (OA/mAcc)|
> |-|-|-|
> |FL-Cosine|83.5/65.3|87.3/74.2|
> |S$^2$P$^2$-Cosine|**84.6**/74.1|**88.6**/80.6|
> |Herding|78.9/51.7|85.1/67.2|
> |S$^2$P$^2$-Herding|**83.2**/71.5|**87.8**/80.1|
> |K-Center|57.9/54.6|83.7/74.1|
> |S$^2$P$^2$-K-Center|64.3/**68.6**|**84.1**/**78.4**|
>
> >**W8.** Limited task diversity.
>
> **A5.** We tested on CIFAR-100-LT (imbalance ratio 10, ResNet-34); S$^2$P$^2$ consistently outperforms all baselines on both OA and mAcc. Full results in A2 of our response to Reviewer bMzp.
>
> >**W9.** Comparisons may not fully test whether proposed principles are necessary.
>
> **A6.** The reviewer raises a valuable point. We compare against three simpler alternatives:
> (1) Balanced subsampling (K=1 in SGS, already in Fig. 7) is suboptimal, as predicted by Thm. 3.3.
> (2) Manually tuned per-class quotas ($\propto n_i$ or $\sqrt{n_i}$) are consistently outperformed by SGS.
> (3) Tab. 4 is already a controlled comparison: all baselines use the same post-selection training as S$^2$P$^2$ (CSL+EGD). The gap is thus attributable to selection, not training.
> Results for (1) and (2):
> ||m=550 (OA/mAcc)|m=2200 (OA/mAcc)|
> |-|-|-|
> |K=1 (balanced)|83.5/75.0|88.5/80.5|
> |Linear quota|84.5/60.2|88.3/69.7|
> |Sqrt quota|84.9/73.8|87.9/81.2|
> |S$^2$P$^2$ (K=0.2)|**85.2**/72.3|**88.9**/79.0|
> |S$^2$P$^2$ (K=0.6)|83.6/**75.3**|87.7/**81.8**|
>
> >**W10.** Claims feel stronger than evidence supports.
>
> **A7.** To our knowledge, we are the first to study dataset pruning in 3D and to identify robustness across target priors as a core requirement. Our design philosophy is to maximize the shared component across priors, with SGS navigating the resulting trade-off. The theoretical grounding has been strengthened (A1,A3). We will ensure claims in the revision reflect these contributions precisely.
>
> >**W11.** Typos, grammar, notation inconsistencies.
>
> **A8.** All issues corrected in the revision.
>
> [1] Golowich et al., Size-Independent Sample Complexity of Neural Networks.
>
> [2] Chen et al., Super-Samples from Kernel Herding.

---

> > ### Author Rebuttal · Reviewer_V9Ex · 2026-04-04
> >
> > Nice. I appreciate the additional work during rebuttal. I have adjusted (increased) my score and do include these revisions and feedback to improve the quality of manuscript.

---

> > > ### Author Response · Authors · 2026-04-04
> > >
> > > We sincerely thank the reviewer for the thoughtful and encouraging comments. We will carefully incorporate all feedback into the revised manuscript to further strengthen the paper. If you have any further questions, we are happy to address them.

---

### Decision · Program_Chairs · 2026-04-30

**Decision:**

Accept (regular)

**Comment:**

This paper studies 3D dataset pruning under class imbalance. Reviewers agreed that this is a relevant and underexplored problem, and found the paper’s main strength to be a coherent prior-aware framework combining a useful conceptual formulation, geometry-based selection, a per-class safety floor, distillation-based training, and a simple steering mechanism. The empirical evaluation was considered broad and convincing, with consistent gains especially on mAcc while remaining competitive on OA.

The main concerns were that the theory is more explanatory than a tight justification of the algorithm, the novelty lies more in the synthesis than in any single new component, and the original presentation and empirical support could be stronger. The rebuttal clarified the theory–method connection, added supporting ablations and multi-seed evidence, and addressed writing issues. These responses substantially resolved the concerns, and all reviewers support acceptance.

The paper is technically sound, well motivated, and likely to be useful to the ICML community, especially those interested in pruning, long-tailed learning, and 3D recognition. For the final version, the authors should integrate the additional clarifications provided in the rebuttal.